# The mosaic memory of large language models

Igor Shilov [1,2], Matthieu Meeus[1,2] & Yves-Alexandre de Montjoye [1] ✉

As Large Language Models (LLMs) become widely adopted, understanding how they learn from, and memorize, training data becomes crucial. Memorization in LLMs is widely assumed to only occur as a result of sequences being repeated in the training data. Instead, we show that LLMs memorize by assembling information from similar sequences, a phenomenon we call mosaic memory. We show major LLMs to exhibit mosaic memory, with fuzzy duplicates contributing to memorization as much as 0.8 of an exact duplicate and even heavily modified sequences contributing substantially to memorization. Despite models displaying significant reasoning capabilities, we somewhat surprisingly show memorization to be predominantly syntactic rather than semantic. We finally show fuzzy duplicates to be ubiquitous in real-world data, untouched by deduplication techniques. In this work, we show memorization to be a complex, mosaic process, with real-world implications for privacy, confidentiality, model utility and evaluation.

Large Language Models (LLMs) are increasingly transforming various aspects of daily life. They drive advancements across industries by automating routine tasks through applications like virtual assistants and customer service bots[1,2]; extracting valuable insights from unstructured data[3]; generating code to support software development[4,5]; and reinventing information retrieval through conversational AI such as ChatGPT[6].

LLMs develop their capabilities by absorbing patterns from massive text datasets, allowing them to both generalize concepts and selectively memorize information. Generalization allows LLMs to reason[7] and apply their knowledge to unseen scenarios, while memorization helps them retain factual information. The interplay between generalization and memorization in LLMs has been widely studied from a model utility perspective, examining how memorization contributes to generalization[8–10], facilitates the encoding of factual knowledge[11,12], and how different data mixtures influence model performance[13].

However, memorization can also be unintended and raise concerns. Indeed, LLMs have been shown to memorize personal or confidential information[14–19], copyright-protected content[20–22], as well as portions of evaluation benchmarks, potentially inflating performance estimates and undermining fair assessment[23–29].

The mechanism by which LLMs memorize training data has predominantly been understood as exact, or verbatim. Indeed, current studies of LLM memorization have shown that a piece of content repeated exactly several times in the training data has a much higher chance of being memorized[22,30–32]. This memorized content could then be extracted by a potential attacker – either exactly[32] or approximately[33,34]. This behavior is often highly undesirable, especially if the extractable content contains private information or is copyright-protected. This understanding has, in turn, led model developers to implement data deduplication techniques to improve model utility[31,35–39], mitigate privacy and confidentiality risks[30], or to decontaminate benchmark data to properly evaluate model capabilities[40–44]. However, the presumed verbatim nature of memorization has led to most mitigation strategies to rely on removing exact repetitions of sequences of text.

In this work, we argue that viewing LLM memorization solely through the lens of exact repetitions in the training data is incorrect. Instead, we posit that LLMs exhibit a behavior we call mosaic memory, which we define as the model's ability to memorize an arbitrary sequence of tokens through exposure to its fuzzy duplicates – fragments of text similar to the original sequence with some tokens missing, replaced, or shuffled. We show that fuzzy duplicates

[1]Department of Computing and Data Science Institute, Imperial College London, London, UK. [2]These authors contributed equally: Igor Shilov, Matthieu Meeus. ✉e-mail: deMontjoye@imperial.ac.uk

contribute to a sequence's memorization to a degree comparable to exact repetitions.

First, we introduce a framework to study the mosaic memory of LLMs. Building on established techniques for analyzing text memorization[22,30], we evaluate the performance of Membership Inference Attacks (MIAs) on canaries—artificially crafted sequences included as part of the training data[14]. We inject a reference canary (100 tokens), along with its fuzzy duplicates, into the training dataset of a target LLM, and measure how the presence of these fuzzy duplicates impacts MIA performance on the reference canary. We measure the memorization of fuzzy duplicates using exact duplicate equivalent $\rho$, the proportion of the memorization impact retained by the fuzzy duplicate compared to an exact one. A value of $\rho = 1$ indicates that a fuzzy duplicate contributes to memorization of the reference canary as much as an exact duplicate, while $\rho = 0$ would indicate that the fuzzy duplicate has no impact on the memorization.

We study four LLMs: GPT-Neo 1.3B (EleutherAI)[45], Gemma-2B (Google)[46], Phi-2 (Microsoft)[47], and Llama-3.2-1B (Meta)[48], and find all of them to exhibit a significant mosaic memory. For example, fuzzy duplicates with 10% of their tokens replaced contribute to the memorization of the canary between $\rho = 0.60$ (Gemma-2B) and $\rho = 0.65$ (GPT-Neo) of an exact duplicate's impact. Moreover, even heavily modified fuzzy duplicates—where 50% of tokens are replaced—still contribute as much as $\rho = 0.13 - 0.19$ of an exact duplicate. Furthermore, we find that mosaic memory is remarkably robust, as the $\rho$ of fuzzy duplicates remains significantly higher than 0, even when separated by noise insertions or when their order is shuffled.

Second, while memorization is not exact, we find, somewhat surprisingly, that it is still predominantly syntactic, rather than semantic. Despite modern LLMs demonstrating strong performance across tasks requiring semantic understanding—such as reasoning, mathematical and scientific problem solving, and multilingual generalization[46,47,49]—we find that memorization is driven mostly by the model's retention of specific overlapping tokens, and not the encoding of the shared underlying meaning across fuzzy duplicates. Fuzzy duplicates created by replacing tokens while preserving semantic meaning indeed are only marginally better memorized than those where semantic meaning is not maintained. Likewise, our results suggest that paraphrased sequences are primarily memorized due to token overlap rather than their shared semantic meaning.

Third, we investigate how widespread fuzzy duplicates are in the real-world, large-scale dataset used for LLM pretraining, SlimPajama[50]. Although this dataset is already deduplicated on the document level, we find that many sequences still have a substantial number of both exact and fuzzy duplicates. To identify these fuzzy duplicates, we use Levenshtein distance, which measures the minimum number of single-character edits required to transform one sequence into another. Levenshtein distance 10, for instance, represents just 10% of tokens edited and corresponds to $\rho$ values between 0.6 and 0.8. Our analysis reveals that an arbitrary sequence with 1000 exact duplicates also has, on average, 3000 more fuzzy duplicates at Levenshtein distance 10 (representing just 10% of tokens edited) and over 20,000 at distance 50. This suggests that traditional deduplication methods are likely insufficient for addressing memorization concerns for LLMs.

Taken together, our results reveal a critical gap in our understanding of how LLMs learn from and memorize training data, whether beneficial or harmful. Rather than solely memorizing exact repetitions, LLMs also retain information across fuzzy duplicates, often to a degree comparable to exact duplicates. This mosaic memory phenomenon fundamentally challenges current assumptions underlying deduplication and privacy protection practices. For example, common industry-standard techniques like removing exact matches from benchmarks (e.g., 13-g overlaps used in GPT-3 evaluations) fail to eliminate fuzzy duplicates, potentially inflating evaluation scores and undermining fair benchmarking. Similarly, deduplication strategies that target exact duplicates (e.g., substring matching of 50 tokens) neglect mosaic memory, allowing memorization of confidential or copyright-protected content through slightly modified sequences. Thus, current practices offer insufficient privacy protection, incomplete benchmark decontamination, and suboptimal data preprocessing, necessitating more nuanced deduplication techniques that account for the mosaic nature of memorization.

## Results

### A metric to study the memorization of fuzzy duplicates

We instantiate a framework to measure how an LLM memorizes fuzzy duplicates, relative to exact duplicates, in its training data. Our goal is to design a metric that is invariant to the absolute memorization level of individual sequences, specifically capturing how fuzzy duplicates influence memorization regardless of how inherently memorizable the base sequence is.

Following prior work, we quantify memorization of a target language model $\mathcal{M}$ with tokenizer $T$ by measuring the performance of MIAs on artificially crafted sequences, known as canaries[14,22,51–55].

We define a set of reference canaries $\{X_{\text{ref}}^i | i = 1, \ldots, C\}$ where each $X_{\text{ref}}^i$ is a synthetically generated sequence containing exactly 100 tokens, i.e., $|T(X_{\text{ref}}^i)| = 100$. Specifically, we follow the method of Meeus et al.[22] and generate synthetic sequences by sampling tokens autoregressively from a reference LLM, setting the temperature $\mathcal{T} = 1$ to mimic realistic text. Temperature controls the randomness of token sampling: $\mathcal{T} = 1$ uses the model's natural probability distribution, producing coherent yet varied text, while higher values yield increasingly random sequences. We additionally vary the canary sampling temperature in Methods to verify that our results hold for canaries across the range from regular text to rarer and unexpected sequences.

To generate fuzzy duplicates, we consider any algorithm $\mathcal{A}$ that systematically modifies each reference canary, e.g., by replacing $R$ tokens within the sequence at random. Formally, $\mathcal{A} : X_{\text{ref}}^i \mapsto \{X_j^i | j = 1, \ldots, n_{\text{dup}}\}$ produces a set of fuzzy duplicates $X_j^i$ for each $X_{\text{ref}}^i$, where we always consider $X_1^i = X_{\text{ref}}^i$. We explore multiple instantiations of $\mathcal{A}$ and compare their impact on memorization.

We instantiate an MIA with member and non-member reference canaries. For each reference canary $X_{\text{ref}}^i$, we flip a fair coin to determine random variable $b_i - \{0, 1\}$. If $b_i = 1$, we inject $X_{\text{ref}}^i$ and its corresponding fuzzy duplicates $\{X_j^i | j = 2, \ldots, n_{\text{dup}}\}$ into a training dataset $D$; otherwise, neither $X_{\text{ref}}^i$ nor its fuzzy duplicates are included. We then consider a pretrained LLM $\mathcal{M}_0$, which we continue pretraining on $D$, yielding target model $\mathcal{M}$. Details on the continued pretraining are provided in Methods.

We then study how $\mathcal{M}$ memorizes the member reference canaries via the corresponding MIA performance. We apply MIAs[15,56,57] to $\mathcal{M}$, computing a membership score $\alpha(X_{\text{ref}}^i)$ for each $X_{\text{ref}}^i$ based on output token probabilities from $\mathcal{M}$. Using these scores and their ground-truth membership labels $b_i$, we compute the receiver operating characteristic area under the curve (ROC AUC), denoted as $\bar{\phi}$.

If an MIA can reliably distinguish members from non-members based on the observed $\mathcal{M}$, the corresponding AUC will be high (closer to 1), indicating that the target model $\mathcal{M}$ substantially memorizes the reference canaries. Prior work has shown that including a member canary multiple times in the training dataset yields higher AUC, indicating stronger memorization[22,30,32]. This exact duplication corresponds to the case where algorithm $\mathcal{A}$ is the identity transformation.

In this work, we study how different forms of fuzzy duplication, obtained by applying various algorithms $\mathcal{A}$, influence the MIA AUC and, consequently, the memorization of the reference canaries. If the MIA AUC does not substantially increase for a given set of fuzzy duplicates, the target LLM does not memorize across such duplicates in a way that reinforces its memorization of the reference canaries. In contrast, when the MIA AUC substantially increases due to the

presence of fuzzy duplicates, the LLM exhibits strong memorization across these variants – an effect we refer to as mosaic memory.

We specifically evaluate how fuzzy duplicates impact memorization relative to exact duplicates. We define the equivalent number of exact duplicates $v_{eq}$ as the number of exact repetitions of $X_{ref}^i$ needed to achieve the same level of memorization (quantified with MIA AUC) as observed for fuzzy duplicates (i.e., $\tilde{\phi}$). This approach aims to capture how fuzzy duplicates influence memorization regardless of how inherently memorizable a sequence is, as the absolute level of memorization in a given setup is known to be highly sensitive to factors such as model characteristics (e.g., number of parameters), training procedures (e.g., learning rate), and properties of the sequences (e.g., length, perplexity)[22,32].

To compute $v_{eq}$, we first compute how exact duplicates are memorized in the same setup. We repeat the same training procedure, each time injecting an increasing number of exact copies ($v \in \{1, ..., n_{dup}\}$) of the reference canary $X_{ref}^i$ into the training dataset ($D_v$). We measure the resulting MIA performance $\phi_v$ reached for each value of exact repetitions $v$. We then determine for a certain set of fuzzy duplicates associated with algorithm $\mathcal{A}$ and an MIA AUC of $\tilde{\phi}$, the equivalent number of exact duplicates $v_{eq}$ as the value of $v$ for which $\tilde{\phi} \approx \phi_{v_{eq}}$. In other words, $v_{eq}$ represents how many exact repetitions would be needed to achieve the same level of memorization as observed for the fuzzy duplicates. The corresponding values of $\phi_v$ and their evolution with $v$ are reported in Fig. 3.

Finally, we compute the exact duplicate equivalent $\rho$ for a single fuzzy duplicate by normalizing $v_{eq}$ by the total number of fuzzy duplicates considered. As such, the value of $\rho$ is independent of the number of fuzzy duplicates and reflects the equivalent number of exact duplicates represented by a single fuzzy duplicate generated by $\mathcal{A}$. We provide additional details for the computation of $\rho$ in Methods.

**Large language models have a mosaic memory**
Out of the many algorithms $\mathcal{A}$ which can be used to construct fuzzy duplicates, we first consider $\mathcal{A}_{replace}$, which constructs fuzzy duplicates of each reference canary $X_{ref}^i$ by replacing $R$ randomly chosen tokens from $T(X_{ref}^i)$. For each fuzzy duplicate $X_j^i$, where $j \in \{2, ..., n_{dup}\}$, we define a modification set $\mathcal{R}_j^i \subset \{1, ..., |T(X_{ref}^i)|\}$ of size $|\mathcal{R}_j^i| = R$, consisting of token positions randomly sampled without replacement. For each token position $r \in \mathcal{R}_j^i$, the original token $T(X_{ref}^i)_r$ is replaced by a different one. For more details, we refer to Methods.

Our results show that across a range of widely used, recently developed LLMs – GPT-Neo[45], Gemma-2[46], Phi-2[47], Llama-3.2[48]–fuzzy duplicates contribute significantly to memorization. Figure 1a illustrates how $\rho$ varies when the number of replacements $R$ made to the fuzzy duplicates increases. We find the exact duplicate equivalent $\rho$ to decrease gradually with the number of replaced tokens $R$, consistently finding $\rho > 0$ until all tokens in the sequence are replaced ($R = 100$). For instance, considering GPT-Neo 1.3B, when $R = 10$ tokens are replaced (out of the 100 in total), the fuzzy duplicates still yield exact duplicate equivalents of $\rho = 0.65$. In practice, this means that having two fuzzy duplicates in the training data with 10% of the original tokens replaced yields higher memorization than reached for one exact repetition. Even for $R = 50$ replaced tokens, i.e., half of the original tokens, $\rho$ is still maintained significantly above zero at 0.19.

The results are furthermore remarkably consistent across a range of widely used LLMs. While the original training data, benchmark performance, and the extent to which each model memorizes likely differs substantially between, for instance, GPT-Neo 1.3B and the more recent Phi-2, both LLMs show similar memorization behavior in the presence of fuzzy duplicates.

In ablation studies provided in Methods, we also show our findings to be consistent across 3 state-of-the-art MIA methodologies[14,56,57], different learning rates, different model sizes, and sampling temperatures used to generate reference canaries. We further consider

slight modifications to $\mathcal{A}_{replace}$, considering multiple ways of selecting the tokens to be replaced and find this to have limited impact. In line with prior work[15,32,33,58], we also study how fuzzy duplicates affect the data extraction risk of the reference canaries. We measure approximate extraction during greedy decoding[33] and find that fuzzy duplicates contribute similarly to extraction risk as to MIA performance.

So far, we have exclusively considered fuzzy duplicates obtained by replacing $R$ tokens from the reference canary ($\mathcal{A}_{replace}$). Using the same framework, we now explore alternative methods for constructing fuzzy duplicates.

First, we consider inserting random tokens at certain positions in the reference canary ($\mathcal{A}_{insert}$). We split the tokenized reference canary $T(X_{ref}^i)$ into $C_n = \frac{|T(X_{ref}^i)|}{n}$ equally sized $n$-grams and insert $X_{insert}$ random tokens between each $n$-gram to generate the fuzzy duplicates. This allows us to assess the robustness of LLM memorization to noise, while preserving the original tokens of the reference canary in the same relative order, across each fuzzy duplicate.

We compare this approach to the baseline scenario, where $n$-grams are randomly and independently distributed in the training data. In this case, the $n$-grams are encountered by the model in isolation (i.e., likely not within the same context window), preventing the model from learning any associations between them. We refer to this as $X_{insert} = \infty$. This baseline reflects how the memorization of individual subsequences contributes to the memorization of the reference canary. Any additional memorization indicates the model's capacity to ignore inserted random tokens and piece together fragments across fuzzy duplicates, disregarding the inserted tokens as irrelevant noise.

Figure 1b shows how LLMs exhibit remarkable robustness in skipping irrelevant tokens. For example, when $X_{insert} = 1$ random token is inserted between all the 20-g in the reference canary, the exact duplicate equivalent $\rho$ still reaches $\rho = 0.83$, which is significantly larger than the lower bound $\rho = 0.48$ at $X_{insert} = \infty$. When $X_{insert} = 10$ tokens are inserted, the memorization decreases, yet remains as high as $\rho = 0.70$. Even in the extreme case where the entire sequence is split into individual tokens ($n = 1$), all separated by $X_{insert} = 1$ token, we find a $\rho$ of 0.21, far exceeding the baseline assumption that such fuzzy duplicates do not contribute to the memorization of the reference canary ($\rho = 0$).

While this remains to be shown, we hypothesize that such robustness derives from the attention mechanism, which could potentially allow the LLM to memorize a connection between tokens even if irrelevant tokens are inserted in between. During training, the LLM would learn to assign low attention scores to inserted tokens, effectively filtering them out as noise. This would lead the model to focus on meaningful patterns across fuzzy duplicates, reinforcing the connections between the original subsequences. When the reference canary's subsequences then reappear together during inference for the MIA, the model reconstructs these relationships, leading to stronger memorization.

Second, we study the memorization of sequences for which the exact token order is no longer preserved. We now obtain the fuzzy duplicates by shuffling the tokens in the reference canary $X_{ref}^i$ to varying degrees ($\mathcal{A}_{shuffle}$).

$\mathcal{A}_{shuffle}$ first partitions a reference canary $T(X_{ref}^i)$ into $C = \frac{|T(X_{ref}^i)|}{n}$ equally sized $n$-grams for $n \in \{2, 5, 10\}$. Fuzzy duplicates are then generated by randomly permuting these $n$-grams while maintaining the original token order within each $n$-gram. From our findings above for $X_{insert} = \infty$, we observe that individual $n$-grams from the reference canary spread across the dataset contribute non-trivially to memorization. $\mathcal{A}_{shuffle}$ controls for this effect, as it maintains the local order of tokens and shuffles $n$-grams rather than individual tokens.

To quantify the degree of permutation, we use the token-level normalized Kendall tau distance ($\tau$)[59]. Kendall tau measures the proportion of token pairs that maintain their relative order in the fuzzy duplicate compared to the reference canary. Given two sequences of

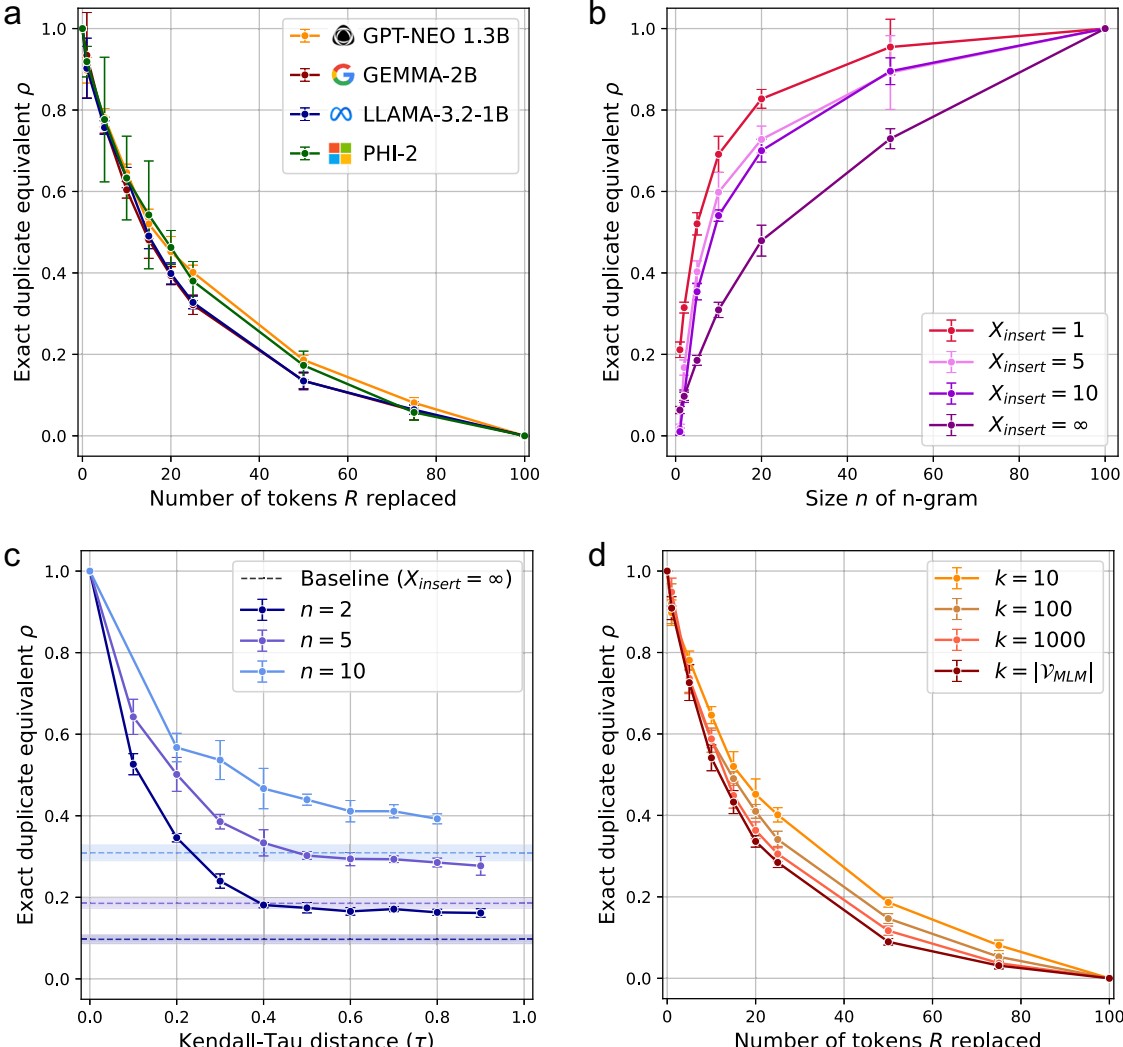

**Fig. 1 | Quantifying how LLMs memorize fuzzy duplicates.** The exact duplicate equivalent $\rho$ (mean and standard deviation over 5 independent runs) for fuzzy duplicates obtained by (**a**) replacing $R$ tokens ($\mathcal{A}_{replace}$), **b** separating $n$-grams by $X_{insert}$ tokens ($\mathcal{A}_{insert}$), **c** shuffling $n$-grams ($\mathcal{A}_{shuffle}$) and **d** replacing $R$ tokens with varying levels of semantic coherence. **a** For smaller values of $R$, fuzzy duplicates contribute to memorization almost as strongly as exact duplicates ($\rho = 1$), while for larger values of $R$, memorization remains significantly higher than if the canary was entirely absent from the training dataset ($\rho > 0$). Results for GPT-Neo 1.3B, Gemma-2B, Phi-2 and Llama-3.2-1B. **b** The model memorizes the canary despite the presence of varying amounts of noise tokens inserted between meaningful chunks. $X_{insert}$ represents different numbers of random tokens inserted between each $n$-gram, with $X_{insert} = \infty$ representing the baseline where $n$-grams are randomly and

independently scattered throughout the training dataset. **c** Results illustrate the impact of token reordering on memorization, with Kendall tau distance ($\tau$) measuring the degree of permutation between token pairs. Higher $\tau$ values indicate greater departure from the original sequence order. Different $n$ values represent different sizes of $n$-grams kept intact while their positions were shuffled. Dashed lines show baseline memorization levels for each $n$ value when $n$-grams are randomly scattered ($X_{insert} = \infty$). **d** Replacement tokens are sampled from the top $k$ predictions returned by masked language model MLM. When $k = 10$, tokens are replaced by one of the most likely 10 tokens predicted by MLM, while for $k = \mathcal{V}_{MLM}$, tokens are replaced by a random token from the MLM's vocabulary. Results for (b-d) are reported for GPT-Neo 1.3B.

tokens, $T(X_{ref}^i)$ (reference canary) and $T(X_j^i)$ (fuzzy duplicate obtained by shuffling), Kendall tau is defined as $\tau = \frac{\Delta}{L(L-1)/2}$, where $L = |T(X_{ref}^i)|$ is the total number of tokens and $\Delta$ is the number of discordant pairs, i.e., token pairs $(t_u, t_v)$ where the relative order is different between $X_{ref}^i$ and $X_j^i$. A value of $\tau = 0$ indicates no shuffling (the original order is preserved), while $\tau = 1$ indicates exact reversal.

For each $n$, we randomly permute $n$-grams to generate fuzzy duplicates and group them based on their Kendall tau distance. Note that since shuffling $n$-grams instead of individual tokens preserves the local order of tokens within the $n$-gram, not all $\tau$ values within $[0, 1]$ can be reached for every $n$.

Figure 1c first reveals that memorization is highly sensitive to ordering. Indeed, we observe that $\rho$ drops from 1 to approximately 0.53 when only 10% of token pairs have their relative order inverted

($\tau = 0.1$), showing that perturbing the sequence even slightly has a significant impact on memorization.

Second, and more interestingly, we find that some information is still memorized even with extensive shuffling. To assess this, we compare the values of $\rho$ obtained at different levels of shuffling to the baseline in which $n$-grams are memorized independently of each other, i.e., when $n$-grams are randomly and independently distributed across the dataset ($X_{insert} = \infty$ in Fig. 1b). Figure 1c then shows that the memorization, across all values $\tau$ remains significantly higher than the baseline $X_{insert} = \infty$. Considering 10-g, for example, we still recover $\rho = 0.44$ (against the baseline of $\rho = 0.31$) even when the order for 50% of the token pairs is inverted ($\tau = 0.5$). These results indicate that the model's mosaic memory retains the shared vocabulary across fuzzy duplicates despite minimal preserved ordering.

**Table 1 | Memorizing paraphrases ($\mathcal{A}_{\text{paraphrase}}$)**

| Fuzzy duplicates | | $\rho$ | n-gram overlap | | |
| --- | --- | --- | --- | --- | --- |
| | | | n = 1 | n = 2 | n = 4 |
| Paraphrased using | Llama-3-8B-Instruct | 0.13 ± 0.01 | 37.15 ± 19.45 | 17.05 ± 15.11 | 7.70 ± 11.15 |
| | Mistral-7B-Instruct | 0.21 ± 0.02 | 56.39 ± 18.51 | 30.97 ± 17.50 | 15.54 ± 14.17 |
| | GPT-4o | 0.34 ± 0.02 | 73.15 ± 16.62 | 48.02 ± 20.58 | 29.44 ± 21.51 |

To obtain the fuzzy duplicates, we paraphrase the reference canaries using three instruction-tuned models (Meta-Llama-3-8B-Instruct, Mistral-7B-Instruct-v0.2, and GPT-4o). We report the corresponding results for the exact duplicate equivalent $\rho$ using GPT-Neo 1.3B as target model, alongside the mean n-gram overlap between the paraphrases and the reference canary. We report the mean and standard deviation aggregated across 5 independent runs. For comparison, we report the results for n-gram overlap for fuzzy duplicates with R replacements in Table 4.

### The mosaic memory is syntactic rather than semantic

All our experiments thus far have focused exclusively on syntactic modifications to create fuzzy duplicates, i.e., systematically altering tokens through replacements, insertions, and shuffling while tracking their impact on memorization. We now examine whether semantic relationships also influence memorization. Our key question is whether LLMs primarily memorize specific tokens or capture underlying meanings. To explore this, we conduct two experiments. First, we systematically vary semantic coherence while controlling token-level changes. Second, we analyze paraphrases as fuzzy duplicates, where semantic meaning is preserved without explicit control over token overlap.

First, we consider the same algorithm to make fuzzy duplicates as above ($\mathcal{A}_{\text{replace}}$), where $R$ tokens are replaced for each fuzzy duplicate $X_j^i$ but now vary the extent to which the replacement maintains the semantic meaning of the sequence. In particular, we leverage a masked language model MLM to predict the top-$k$ tokens (from the MLM's vocabulary $\mathcal{V}_{\text{MLM}}$) for each replacement and vary the value of $k$, thereby changing how semantically meaningful the replacement is. We thus consider algorithm $\mathcal{A}_{\text{replace},k}$ where smaller values of $k$ lead to more semantic similarity between the reference canary and the fuzzy duplicates. We use RoBERTa[60] as MLM, which has a vocabulary size of $|\mathcal{V}_{\text{MLM}}| = 50,265$. For implementation details and examples of token replacements for varying values of $k$, we refer to Methods.

Figure 1d shows the impact of making $R$ replacements with semantically meaningful tokens (smaller $k$) rather than random ones ($k = |\mathcal{V}_{\text{MLM}}|$). While lower values of $k$ consistently lead to higher memorization, the impact is notably small compared to the impact of syntactic changes.

For example, when $R = 10$ tokens are replaced in the fuzzy duplicates, using semantically similar tokens ($k = 10$) only increases $\rho$ by 0.11 (0.54 → 0.65) compared to using a random token ($k = |\mathcal{V}_{\text{MLM}}|$). This effect size is smaller than replacing just 5 more tokens while keeping the semantic meaning intact (for $R = 15$ and $k = 10$, we get $\rho = 0.52$).

These results suggest that the mosaic memory of LLMs is more syntactic—the model memorizes the connection between specific, overlapping tokens across the fuzzy duplicates—than it is semantic, where the model memorizes the underlying semantic meaning shared by all fuzzy duplicates. This finding is somewhat surprising in light of the increasingly strong semantic capabilities demonstrated by modern LLMs, including multi-step reasoning, mathematical and scientific problem solving, code generation, instruction following, and multilingual generalization[46,47,49], which are widely interpreted as relying on abstract semantic representations rather than surface-level token statistics. The fact that memorization nevertheless appears to be driven predominantly by token overlap suggests that, at least for memorization mechanisms, these models may still rely heavily on shallow statistical regularities[61]. We note that the smaller scale of our models and the use of continued rather than full pretraining may influence the degree of semantic memorization observed. Future research is needed to quantify whether the balance between syntactic and semantic memorization differs for larger models trained from scratch.

We further investigate the memorization of fuzzy duplicates designed to maintain semantic coherence, while not explicitly controlling the token overlap. Indeed, algorithm $\mathcal{A}_{\text{replace},k}$ is constrained by only replacing a fixed set of tokens, which might not give sufficient flexibility to maintain semantic coherence using just MLM replacements. Hence, we now study $\mathcal{A}_{\text{paraphrase}}$, which asks an instruction-tuned LLM to paraphrase the reference canary to construct fuzzy duplicates. We consider widely used instruction-tuned LLMs: Meta-Llama-3-8B-Instruct[49], Mistral-7B-Instruct-v0.2[62] and GPT-4o[63], with additional details provided in Methods.

Table 1 shows that for all paraphrases generated using the 3 instruction-tuned LLMs, the exact duplicate equivalent $\rho$ remains fairly low, ranging from 0.13 for Llama-3-8B to 0.34 for GPT-4o[63]. Moreover, this is remarkably low compared to the memorization achieved when $R$ tokens are replaced with random tokens ($k = |\mathcal{V}_{\text{MLM}}|$) from Fig. 1(d). For instance, fuzzy duplicates with 15% of all tokens in the reference canary ($R = 15$) replaced by random ones—the semantic meaning is significantly distorted—still contribute more to the memorization of the reference canary ($\rho = 0.43$) than any of the paraphrased fuzzy duplicates. Only when making $R = 20$ replacements with random tokens, the level of memorization ($\rho = 0.34$) becomes similar to the largest one achieved using paraphrasing ($\rho = 0.34$).

With $\rho$ for paraphrases already relatively low, we hypothesize that the memorization we observe in this experiment can be explained mostly by syntactic similarity, as paraphrases might still have an overlap in tokens with the reference canary. To investigate this, we compute the mean number of overlapping n-grams between each reference canary and its paraphrases for different values of $n$, reporting the results in Table 1. We show that, across paraphrases, there remains a substantial n-gram (i.e., syntactic) overlap, up to an average of 73 1-g present in paraphrases made by GPT-4o and that the memorization strictly increases with n-gram overlap. Specifically, as the mean overlap in 4-g increases from 7.7 (Llama-3-8B) to 15.5 (Mistral-7B) and 29.4 (GPT-4o), the respective $\rho$ also rises from 0.13 to 0.21 and 0.34, respectively—a trend that holds across different values of $n$. While these results do not rule out that some semantic memorization might happen, taken together, they suggest that it likely plays a limited role compared to syntactic memorization.

### Fuzzy duplicates are widespread and robust to deduplication

Having demonstrated the significant contribution of fuzzy duplicates to model memorization, we now investigate their prevalence in a real-world training dataset. Our analysis shows that even in an extensively deduplicated dataset, a large number of fuzzy duplicates exist, significantly contributing to memorization.

To identify fuzzy duplicates in real-world datasets, we adopt Levenshtein distance, which measures the minimum number of single-character edits required to transform one sequence into another. It accounts for token replacements ($\mathcal{A}_{\text{replace}}$), insertions ($\mathcal{A}_{\text{insert}}$), and effectively captures token shuffling ($\mathcal{A}_{\text{shuffle}}$) through combinations of deletion and addition operations.

We validate this choice by investigating the relationship between Levenshtein distance and the exact duplicate equivalent ($\rho$) across our

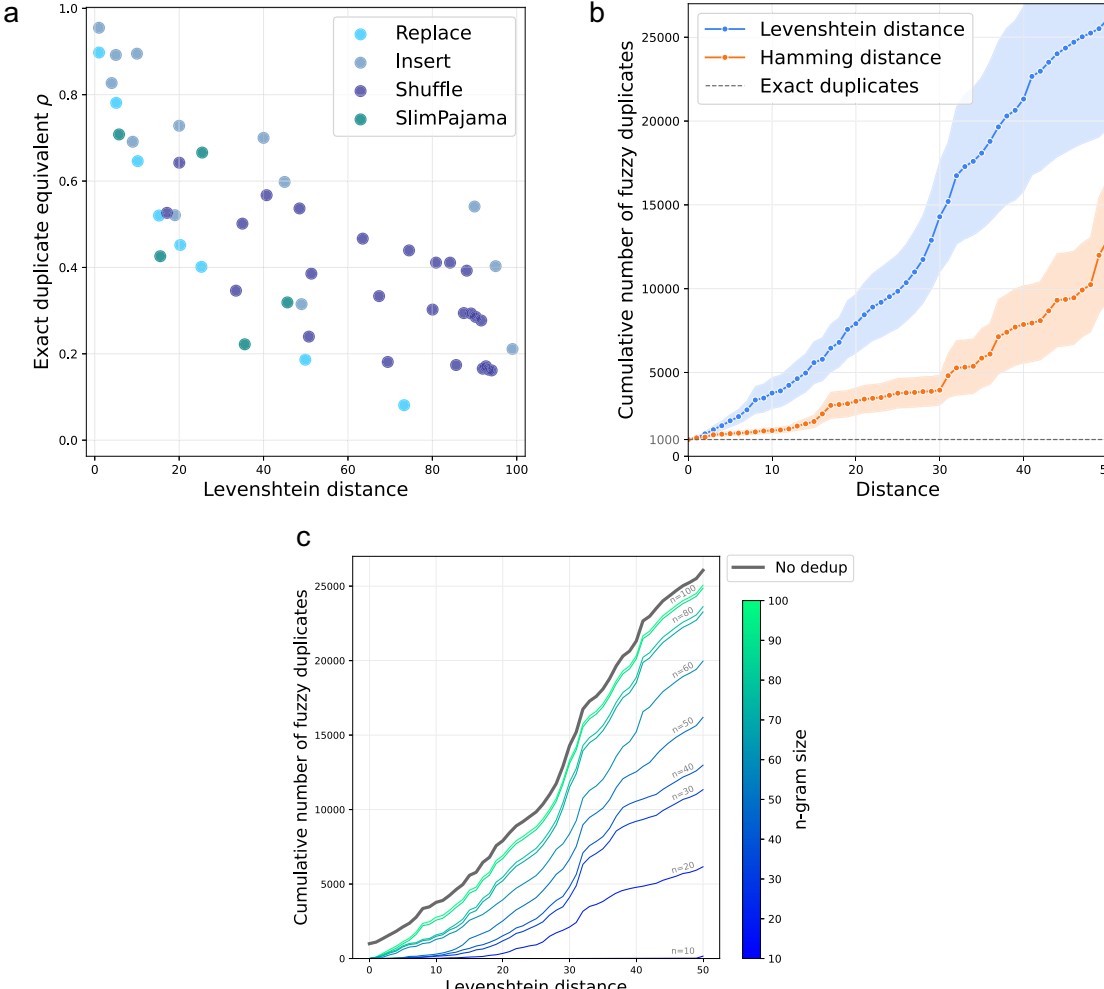

**Fig. 2 | Fuzzy duplicates in SlimPajama. a** We visualize the relationship between the mean Levenshtein distance between fuzzy duplicates and their reference canaries and the corresponding exact duplicate equivalent $\rho$ based on the experimental results reported for $\mathcal{A}_{\text{replace}}$, $\mathcal{A}_{\text{insert}}$ and $\mathcal{A}_{\text{shuffle}}$. For comparison, we also report results for real-world fuzzy duplicates extracted from the SlimPajama dataset. **b** Number of fuzzy duplicates (cumulative) found in SlimPajama with increasing Hamming and Levenshtein distance from the original sequence. Reported numbers are averaged over 100 sequences from a subgroup of sequences repeated verbatim 1000 ( ± 1%) times in the dataset. Shaded regions represent the standard error of the mean. **c** The number of fuzzy duplicates impacted by a varying level of deduplication. $n$-gram deduplication strategies for varying $n$ fail to account for a range of fuzzy duplicates still contributing substantially to memorization.

experimental results. Figure 2a shows a strong negative correlation between $\rho$ and Levenshtein distance, confirming that this metric effectively captures the memorization impact of various fuzzy duplicates.

We also investigate Hamming distance, which counts the number of positions where corresponding tokens differ between two sequences. This metric directly corresponds to $\mathcal{A}_{\text{replace}}$, making fuzzy duplicates identified by Hamming distance real-world examples of our token replacement experiments.

As a target dataset we consider SlimPajama[50]—a thoroughly cleaned and document-level deduplicated dataset derived from the RedPajama dataset[64], containing 627 billion tokens, or 895 GB of text in a compressed form.

The dataset's size adds additional complexity to the analysis, as computing Levenshtein distance for all possible sequence pairs would be computationally infeasible. Given these constraints, we implement a sampling strategy by identifying a small representative set of sequences and scanning the dataset linearly to find their fuzzy duplicates. For computational reasons, we also only scan 5% of the dataset for fuzzy duplicates, and extrapolate the results assuming the uniform distribution of fuzzy duplicate positions—a reasonable assumption given the dataset is randomly shuffled.

For the target sequence selection we focus on those sequences which are repeated exactly multiple times in the dataset. We do not expect randomly chosen sequences to have many fuzzy duplicates, nor is it necessary to establish our point. Our aim is simply to demonstrate that fuzzy duplicates are sufficiently prevalent to raise memorization concerns. We first confirm that our selection criteria represents a substantial population within the dataset, not just isolated cases. We then analyze these sequences to demonstrate the prevalence of their fuzzy duplicates. If positive, these results would establish the existence of fuzzy duplicates in practice. Note that, however, a thorough quantification of the issue is still required, considering the specifics of the desired objective: privacy and confidentiality protections, benchmark decontamination procedures, or overall model utility considerations.

We hypothesize that the number of fuzzy duplicates correlates with the number of exact repetitions in the dataset. Consequently, we select 100 target sequences that are repeated exactly 1000 ( ± 1%) times in SlimPajama and scan for their fuzzy duplicates. We confirm this selection to be meaningful as there are over 700,000 sequences in the dataset repeated at least 1000 times. It makes such sequences sufficiently prevalent so that, if memorized, they would have a major impact on the model's behavior. Implementation details and results

from additional repetition frequency buckets (100 and 10,000) are provided in Methods.

Figure 2b shows how fuzzy duplicates in SlimPajama increase sharply with Levenshtein distance. Beyond our baseline of 1000 exact duplicates (which we explicitly selected for), we find approximately 5000 sequences at Levenshtein distance ≤14, representing 4000 additional fuzzy duplicates. At this distance range, our experiments show exact duplicate equivalent $\rho$ to be between 0.6 and 1.0, indicating these fuzzy duplicates significantly contribute to memorization, with their cumulative impact likely to outweigh that of exact repetitions.

As we extend to larger Levenshtein distances up to 50, corresponding to $\rho$ between 0.2 and 0.4, the number of fuzzy duplicates expands to over 25,000—more than 25 times the original count of exact duplicates. Even when using the more conservative Hamming distance metric, which directly corresponds to the token replacements ($R$) in our $\mathcal{A}_{\text{replace}}$ algorithm and provides a direct real-world analog to our controlled canary experiments with predictable memorization effects, we still find approximately 3000 fuzzy duplicates at distance 20 and 8000 at distance 40.

To illustrate the mechanisms by which numerous fuzzy duplicates arise in real datasets, we provide representative examples in Supplementary Table 4. A notable case is a series of Wikipedia articles describing French communes, which follow an almost identical template where only the commune name and a few key attributes are changed.

Lastly, we validate that our results from previous sections generalize to real-world fuzzy duplicates. Specifically, we select a set of reference canaries and fuzzy duplicates identified in the SlimPajama dataset and repeat the computation of the exact duplicate equivalent $\rho$. The resulting values are shown in Fig. 2a. We observe that, for these real-world fuzzy duplicates, $\rho$ similarly decreases with increasing Levenshtein distance, consistent with the trend observed in our synthetic experiments. Further experimental details are provided in Methods.

We further examine how commonly deployed deduplication practices remove fuzzy duplicates. SlimPajama[50] has been obtained by deduplicating RedPajama[64], removing documents with a Jaccard similarity based on 13-g higher than 0.8 (resulting in the removal of 49.6% of all bytes). Document-level deduplication has been shown to result in a dataset with higher information density[50], allowing developers to train better models with less computational cost[31]. Yet, despite this extensive document-level deduplication, our results showed that SlimPajama still contains a substantial number of exact and fuzzy (sequence-level) duplicates, as document-level deduplication only removes near-identical documents while allowing overlapping sequences to persist across them.

Some model developers have also adopted sequence-level deduplication – a more expensive, yet more granular alternative to document-level deduplication. Following Lee et al.[31], a common approach is to remove exactly duplicated substrings of at least 50 tokens from the training data[30,65,66]. Research shows duplicate sequences in training data increase susceptibility to privacy attacks[22,32], making sequence-level deduplication (typically at 50 tokens) an important mitigation strategy[30]. Despite its benefits for dataset efficiency and mitigating privacy risks, sequence-level deduplication requires a more careful calibration than document-level deduplication. Removing sequences may disrupt the internal coherence of documents, and aggressive sequence-level deduplication risks removing a substantial amount of valuable data.

We examine how sequence-level deduplication affects the exact and fuzzy duplicates present in SlimPajama. Specifically, we consider a fuzzy duplicate removed by sequence-level deduplication, if it shares at least one overlapping $n$-gram with the target sequence. While we note that fuzzy duplicates could also be removed due to an overlap with other sequences in the overall dataset, deduplicating the entire

dataset comes at a high computational cost. We approximate its effect by considering the set of fuzzy duplicates for increasing Levenshtein distances from Fig. 2b and argue that the most significant impact is captured by considering this set of highly similar sequences.

Figure 2c shows the number of fuzzy duplicates that remain after sequence-level deduplication for varying values of $n$. We first examine $n = 50$, a common choice in training data preprocessing[30,31,65,66] or to mitigate privacy risks[30]. While deduplicating with $n = 50$ removes a substantial amount of fuzzy duplicates—particularly at low Levenshtein distances (≤10), a significant number remain. For instance, at a Levenshtein distance of 20 an average of 2500 fuzzy duplicates per sequence persist after deduplication, increasing to 6000 for a Levenshtein distance of 30. At the same time, fuzzy duplicates of similar Levenshtein distances still contribute substantially to memorization ($\rho > 0.3$ in Fig. 2a). These results suggest that standard 50-g deduplication likely fails to properly mitigate any privacy risks and might be suboptimal to enhance training efficiency and model performance. While more aggressive sequence-level deduplication (e.g., $n = 20$) might help remove the most impactful fuzzy duplicates, it likely introduces significant trade-offs.

## Discussion

The mosaic memory of LLMs that we here thoroughly study has important implications. We distinguish between impact on (i) privacy and confidentiality, (ii) deduplication to improve model utility and training efficiency, (iii) benchmark decontamination, (iv) (adversarial) canaries and (v) machine unlearning.

LLMs are trained on extensive corpora of text, typically involving a pretraining stage using vast amounts of publicly available internet data, followed by a posttraining stage to enhance instruction-following capabilities, reduce harmful outputs, or develop domain-specific expertise[43,67,68]. During posttraining, models may be trained on specialized datasets, such as healthcare records[69,70] or human conversations[71], which can introduce personal or sensitive information. Even during pretraining, LLM developers increasingly collect valuable data through licensing deals with, for instance, academic or news publishers[72,73]—data which may be publicly available but is not necessarily free of sensitive or legally protected information. At the same time, LLMs have been shown to reproduce specific sequences verbatim[14,15] and be vulnerable to other privacy attacks such as MIAs[14,56,57,74].

To mitigate privacy risks, training data can be sanitized by removing Personally Identifiable Information (PII) typically using Named Entity Recognition (NER)[75]. NER is, however, known to be an imperfect process[76], with what constitutes PII being context-dependent[16] and models trained on sanitized data remain vulnerable to privacy attacks[18,77]. Regarding confidential information, likely not directly linked to individuals and thus not picked up by NER, distinguishing what needs to be removed from what benefits utility is often even more context-dependent, subjective, and thus a difficult task.

Sequence-level deduplication has also been applied as a mitigation strategy[30], as the risk of privacy attacks increases when more exact repetitions of sequences appear in the training data[22,32]. However, our results show that removing all exact occurrences of 50 tokens might not be sufficient. Indeed, in this work, we study the memorization of fuzzy duplicates by how they contribute to the susceptibility to MIAs, finding that failing to account for them substantially increases the risk. As protection against MIAs also implies protection against reconstruction and inference attacks[78], we further confirm this impact of fuzzy duplicates to generalize to extraction attacks[15,32,33,58] in Methods.

Our findings suggest that currently deployed deduplication techniques relying on exact $n$-gram matching for $n = 50$[30,31,65,66] leave many fuzzy duplicates intact. As a result, deduplicated training data might still contain substantial redundancies, implying suboptimal data preprocessing. More advanced deduplication techniques, for instance

based on Levenshtein or Hamming distance, could further improve training efficiency, achieving comparable or superior performance with less data and lower cost.

Ultimately, the right deduplication technique to reach optimal training efficiency or model utility involves challenging trade-offs that must be carefully considered by model developers. For instance, computing Levenshtein distance across an entire dataset is more expensive than using Hamming distance or exact deduplication. Similarly, determining a deduplication threshold (e.g., value of $n$ in exact deduplication or a maximum distance between sequences) requires balancing data retention with redundancy reduction.

Further work has also proposed semantic deduplication[79] – removing fuzzy duplicates based on their similarity in a semantically meaningful embedding space computed using pretrained models. They report efficiency gains up to 15% for smaller language models trained on semantically deduplicated datasets, outperforming exact deduplication[31]. Such semantic deduplication likely comes with similar trade-offs as discussed above. Interestingly, our findings suggest that LLMs tend to memorize more strongly based on syntactic rather than semantic similarity. We hence hypothesize that more aggressive syntactic deduplication strategies may be even more effective than semantic approaches for improving training efficiency.

Recent research indicates that the performance reported for newly released LLMs may be inflated and misrepresent progress, as models might be trained on and memorize the benchmarks used to evaluate them[23–29,80]. To mitigate this, sequence-level deduplication techniques based on $n$-gram matching are used for benchmark decontamination. Benchmark samples containing $n$-grams overlapping with the training data are removed, and model performance is computed on the remaining samples. This approach was first introduced to evaluate GPT-3[40] detecting an overlap of an equivalent of $n = 25$ tokens and has since been applied to other models[41,42]. This level of deduplication is notably more aggressive than what is commonly used for training data preprocessing ($n = 50$).

Figure 2c shows, however, that many fuzzy duplicates contributing to memorization still remain untouched for $n = 20$, which we use as a conservative approximation of benchmark decontamination. Specifically, we recover thousands of fuzzy duplicates at Levenshtein distances of 20–50, where the exact duplicate equivalent remains $\rho \geq 0.2$ (Fig. 2a). This suggests that the currently deployed decontamination techniques might not be as effective as believed to properly clean benchmarks and ensure the fair evaluation of LLMs, including their reasoning capabilities.

Subsequent work has explored other techniques for benchmark decontamination. PaLM[44], for instance, removes samples with at least 70% overlap in 8-g, while GPT-4[67] excludes those for which any of 3 random sequences of 50 characters overlap. Llama-2[43] filters samples where over 20% of the tokens appear in 10-g (at the token level) from the training data, while allowing a "skipgram budget" of 4 tokens. Finally, a more recent approach suggests determining the decontamination threshold for which model performance diverges the most[81], as used to evaluate Llama-3[48]. While these methods do not directly map to specific Levenshtein distances, our findings suggest that exact $n$-gram deduplication alone fails to eliminate all fuzzy duplicates. More granular approaches, like those used for Llama-3[48], might be more effective, yet have so far not been adopted more widely.

Memorization of canaries has also been used as a tool to infer whether certain content was used to train an LLM, e.g., to detect copyright violations or whether an LLM was trained on an evaluation benchmark. Indeed, it has been proposed to include exact repetitions of unique sequences–either random token sequences[82] or synthetically generated copyright traps[22], as part of original content. Performing an MIA on these sequences then provides a means of detecting potential copyright violations during model training. Similarly, a unique GUID string was added to the BIG-bench evaluation benchmark[83], both to facilitate model developers to filter out benchmark files from their training data and to enable post-hoc verification of whether an LLM was trained on the benchmark.

However, exact deduplication would likely remove such canaries. For instance, copyright traps only yield meaningful memorization for 100-token sequences repeated 1000 times[22], which would be trivially removed by commonly deployed deduplication techniques.

Our findings suggest that the mosaic memory of an LLM could be leveraged to design canaries that are resistant to training data deduplication practices while still being meaningfully memorized by LLMs. Similar techniques could also be leveraged to induce memorization of specific content, including biased opinions or misinformation.

Lastly, the mosaic memory of LLMs introduces additional challenges for machine unlearning. Unlearning refers to post hoc methods that make a model forget part of its training data when retraining from scratch is prohibitively expensive. Motivations include privacy concerns and regulatory requirements such as the GDPR's "right to be forgotten"[84,85], intellectual property obligations when models memorize copyrighted data[86–88], and safety considerations such as reducing harmful knowledge[89,90].

Most unlearning techniques, particularly those motivated by privacy regulation, rely on a forget set–a predefined list of training samples to be removed[91,92]. Their effectiveness is typically evaluated on exact members of that set, yet models have been shown to retain information that can be recovered when the same content is queried in a slightly altered form[93,94].

Our findings show that mosaic memory exacerbates this limitation: because LLMs memorize across clusters of fuzzy duplicates, forgetting an exact instance might not eliminate the underlying knowledge. Effective unlearning will therefore require methods that act on groups of correlated examples rather than exact, isolated points.

## Methods

### Reference canaries

We construct a reference canary $X_{\text{ref}}^i$ by synthetically generating a sequence using a reference language model $\mathcal{M}_{\text{ref}}$ following the approach of Meeus et al.[22]. Specifically, starting from an empty string, we iteratively sample the next token from $\mathcal{M}_{\text{ref}}$'s predicted probability distribution, using temperature $\mathcal{T}$. We further control for the length of the sequence by truncating the synthetically generated sequence to the first $L_{\text{ref}}$ tokens using the tokenizer $T$ of the target model. We query $\mathcal{M}_{\text{ref}}$ for synthetic sequences repeatedly and apply rejection sampling until we have a sufficient number of reference canaries satisfying $|T(X_{\text{ref}}^i)| = L_{\text{ref}}$.

As reference model $\mathcal{M}_{\text{ref}}$, we use the pretrained Llama-2 7B[43] to generate $C = 200$ synthetic reference canaries (100 members and 100 non-members), with length $L_{\text{ref}} = 100$. Unless stated otherwise, we use the default temperature $\mathcal{T} = 1.0$ to sample from $\mathcal{M}_{\text{ref}}$, which we empirically find to lead to meaningful sequences. Only for the results in Fig. 3(a), we use other reference canaries generated using $\mathcal{T} = 2.5$ and $\mathcal{T} = 5.0$. We provide example reference canaries generated with varying sampling temperature in Supplementary Table 1.

Throughout this work, we report the mean and standard deviation of the exact duplicate equivalent $\rho$ (e.g., in Fig. 1a–d). These values are obtained from 5 independent runs, each considering a distinct set of reference canaries generated using different random seeds.

### Generating fuzzy duplicates

To generate fuzzy duplicates, we consider any algorithm $\mathcal{A}: X_{\text{ref}}^i \mapsto \{X_j^i | j = 1, \ldots, n_{\text{dup}}\}$ which systematically modifies reference canary $X_{\text{ref}}^i$ to construct a set of fuzzy duplicates $X_j^i$. We always ensure $X_1^i = X_{\text{ref}}^i$ and use $n_{\text{dup}} = 10$. We consider a range of algorithms $\mathcal{A}$ throughout this work, and formalize each of them below.

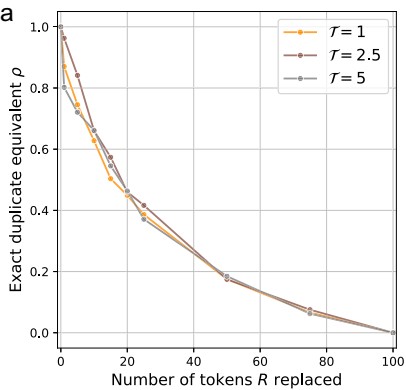
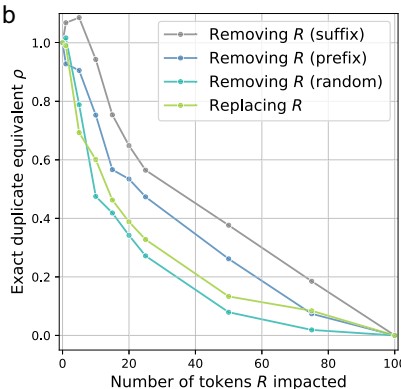
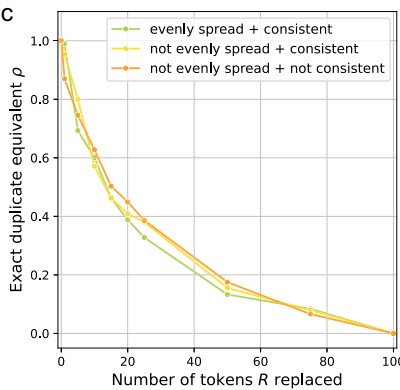

**Fig. 3 | Ablation studies for fuzzy duplicate generation. a** The exact duplicate equivalent $\rho$ for fuzzy duplicates across number of replacements made, considering different values of the temperature $\mathcal{T}$ used to synthetically generate reference canaries using Llama-2 7B as reference model $\mathcal{M}_{\text{ref}}$. **b** The exact duplicate equivalent $\rho$ for fuzzy duplicates when tokens are removed (randomly or from the prefix or suffix) or replaced with a semantically meaningful token. **c** The exact duplicate equivalent $\rho$ for fuzzy duplicates when tokens are replaced with a semantically meaningful token, varying which tokens in the reference canary are replaced. All results in (**a–c**) are computed using GPT-Neo 1.3B.

First, we formalize algorithm $\mathcal{A}_{\text{replace}}$, which constructs fuzzy duplicates by replacing $R$ tokens. Throughout many experiments in this work (e.g., Fig. 1a), we make $R$ token replacements to construct the fuzzy duplicates from the tokenized reference canary $T(X^i_{\text{ref}})$, consisting of exactly $L_{\text{ref}}$ tokens.

For each fuzzy duplicate $X^i_j$, where $j \in \{2, \ldots, n_{\text{dup}}\}$, we define a modification set $\mathcal{R}^i_j \subset \{1, \ldots, |T(X^i_{\text{ref}})|\}$ of size $|\mathcal{R}^i_j| = R$, consisting of token positions randomly sampled without replacement. To make token replacements, we consider replacing $R \in \{1, 5, 10, 15, 20, 25, 50, 75\}$ of the 100 tokens in the tokenized reference canary.

For each token position $r \in \mathcal{R}^i_j$, the original token $T(X^i_{\text{ref}})_r$ is replaced by a new token sampled from the top-$k$ most probable tokens predicted by the masked language model MLM. Specifically, given the masked sequence $T(X^i_{\text{ref}})_{\backslash r}$, where the token at position $r$ is removed, MLM assigns a probability distribution $P_{\text{MLM}}(\cdot|T(X^i_{\text{ref}})_{\backslash r})$ over the vocabulary. The replacement token is then drawn uniformly at random from the top-$k$ most probable tokens according to $P_{\text{MLM}}$. Thus, each fuzzy duplicate $X^i_j$ is generated as:

$$T(X^i_j)_r \sim \text{Top-}k\left(P_{\text{MLM}}(\cdot|T(X^i_{\text{ref}})_{\backslash r})\right), \forall r \in \mathcal{R}^i_j. \quad \text{(M1)}$$

Note that the MLM tokenizer's vocabulary likely does not align with the vocabulary of tokenizer $T$, which defines the sequence length of $L_{\text{ref}} = 100$ tokens. A single replacement token sampled from the MLM vocabulary may thus correspond to multiple tokens under tokenizer $T$. In practice, we obtain exactly one replacement token from the MLM's vocabulary $\mathcal{V}_{\text{MLM}}$, decode this to text, and then re-tokenize this text using $T$ before inserting it back into the sequence. As a result, the length of a fuzzy duplicate, measured in tokens under $T$, may slightly exceed $L_{\text{ref}}$.

Further, when $k$ equals the size of the entire vocabulary $\mathcal{V}_{\text{MLM}}$ of tokenizer $T_{\text{MLM}}$, or $k = |\mathcal{V}_{\text{MLM}}|$, we replace the token with a randomly sampled token from $\mathcal{V}_{\text{MLM}}$. This likely distorts the fluency and semantic meaning of the sequence. For smaller values of $k$, the token is replaced by a token which corresponds to a higher predicted probability returned by MLM, likely maintaining the fluency and semantic meaning of the sequence. We use RoBERTa[60] as MLM, which has a vocabulary size of $|\mathcal{V}_{\text{MLM}}| = 50,265$. We provide example fuzzy duplicates for varying $k$ in Supplementary Table 2.

Lastly, to generate the fuzzy duplicates using $\mathcal{A}_{\text{replace}}$, we can additionally vary which tokens from the original reference canary are replaced. Specifically, we define the modification sets $\mathcal{R}^i_j \subset \{1, \ldots, |T(X^i_{\text{ref}})|\}$, where each $\mathcal{R}^i_j$ consists of $R$ token positions subject to

different selection strategies. We distinguish between the following cases:

1. "Evenly spread + consistent": The $R$ replacement positions are chosen to be evenly distributed across the reference canary. The sequence is partitioned into $R$ contiguous segments of (approximately) equal size, and one token position is selected at random from each segment. Formally, we define

$$S_i = \{S_{i,1}, S_{i,2}, \ldots, S_{i,R}\}, \text{ where } S_{i,r} = \left[\frac{(r-1)}{R}|T(X^i_{\text{ref}})|, \frac{r}{R}|T(X^i_{\text{ref}})|\right) \quad \text{(M2)}$$

   as the $r$-th segment of token indices. The global modification set $\mathcal{R}_i$ is then sampled as $\mathcal{R}_i = \{r_{i,1}, r_{i,2}, \ldots, r_{i,R}\}$, where $r_{i,r} \sim S_{i,r}$. This set is used consistently across all fuzzy duplicates, i.e., $\mathcal{R}^i_j = \mathcal{R}_i, \forall j \in \{2, \ldots, n_{\text{dup}}\}$.

2. "Not evenly spread + consistent": The $R$ replacement positions are sampled uniformly at random over token positions, forming one global modification set $\mathcal{R}_i$. This set is then used consistently across all fuzzy duplicates, i.e., $\mathcal{R}^i_j = \mathcal{R}_i, \forall j \in \{2, \ldots, n_{\text{dup}}\}$.

3. "Not evenly spread + not consistent": The $R$ replacement positions are sampled uniformly at random over token positions, but independently for each fuzzy duplicate. That is, for each duplicate $X^i_j$, we independently sample $\mathcal{R}^i_j$ for each $j \in \{2, \ldots, n_{\text{dup}}\}$. This results in different token replacement sets across fuzzy duplicates.

Throughout this work, unless specified otherwise, we construct fuzzy duplicates using $\mathcal{A}_{\text{replace}}$ which we consider the most realistic to appear in real-world datasets. For this, we replace tokens with a semantically meaningful other token (opting for $k = 10$), not uniformly spread across the reference canary and not consistent across fuzzy duplicates ("not evenly spread + not consistent"). We study multiple values of $k$ in Fig. 1d and different ways of replacing tokens across fuzzy duplicates in Fig. 3c.

Next, we formalize algorithm $\mathcal{A}_{\text{insert}}$, which constructs fuzzy duplicates by inserting random tokens. For each reference sequence $X^i_{\text{ref}}$, we first partition its tokenized form $T(X^i_{\text{ref}})$ into $C_n$ contiguous $n$-grams, where $C_n = \frac{|T(X^i_{\text{ref}})|}{n}$. To generate fuzzy duplicates, we insert $X_{\text{insert}}$ tokens between each $n$-gram. The inserted tokens are sampled independently from the vocabulary of the target model, or $I_c \sim \text{Uniform}(\mathcal{V}_{\mathcal{M}})^{X_{\text{insert}}}$. The fuzzy duplicate is then constructed as

$$T(X^i_j) = \bigcup_{c=1}^{C_n-1}\left(T(X^i_{\text{ref}})_c \cup I_c\right). \quad \text{(M3)}$$

Insertions are resampled for each fuzzy duplicate $X_j^i$, for $j \in \{2, \ldots, n_{dup}\}$. We study the memorization of fuzzy duplicates constructed using $\mathcal{A}_{insert}$ in Fig. 1b.

Lastly, we formalize algorithm $\mathcal{A}_{shuffle}$, which constructs fuzzy duplicates by shuffling tokens. For each reference sequence $X_{ref}^i$, we again partition its tokenized form $T(X_{ref}^i)$ into $C_n$ contiguous $n$-grams, where $C_n = \frac{|T(X_{ref}^i)|}{n}$. To generate fuzzy duplicates with varying degrees of token order permutation, $\mathcal{A}_{shuffle}$ randomly swaps these $n$-grams while maintaining the original token order within each $n$-gram.

Formally, let $G = \{g_1, g_2, \ldots, g_{C_n}\}$ represent the sequence of $n$-grams from the tokenized reference canary, where each $g_j$ is an ordered sequence of $n$ consecutive tokens. $\mathcal{A}_{shuffle}$ generates a fuzzy duplicate $X_j^i$ by creating a permutation $\pi$ of indices $\{1, 2, \ldots, C_n\}$ and concatenating the $n$-grams in the permuted order: $T(X_j^i) = g_{\pi(1)} \circ g_{\pi(2)} \circ \ldots \circ g_{\pi(C_n)}$.

To quantify the degree of permutation between the reference canary and each fuzzy duplicate, we use the normalized Kendall tau distance $\tau$, which measures the proportion of token pairs whose relative order differs between the two sequences. For token positions $u$ and $v$, let $(t_u, t_v)$ represent a token pair in the reference canary. This pair is considered discordant if its relative order in the fuzzy duplicate is inverted. The Kendall tau distance is defined as:

$$\tau = \frac{\Delta}{L(L-1)/2} \tag{M4}$$

where $L = |T(X_{ref}^i)|$ is the total number of tokens and $\Delta$ is the number of discordant pairs.

For our experiments, we generate fuzzy duplicates through rejection sampling, continuously permuting $n$-grams and computing $\tau$ until we have obtained the desired number of fuzzy duplicates for a given Kendall tau distance. We study the memorization of fuzzy duplicates constructed using $\mathcal{A}_{shuffle}$ in Fig. 1c.

### Membership inference game

Throughout this work, we measure memorization of a target language model $\mathcal{M}$ with tokenizer $T$ by instantiating a membership inference attack (MIA) on the artificially crafted reference canaries. We consider $C = 200$ reference canaries $\{X_{ref}^i | i = 1, \ldots, C\}$ with their fuzzy duplicates $\{X_j^i | j = 2, \ldots, n_{dup}\}$, always considering $X_1^i$ equal to $X_{ref}^i$.

We apply continued pretraining on $\mathcal{M}_0$ using the training dataset $D$, denoting the resulting model as the target model $\mathcal{M}$. For each reference canary $X_{ref}^i$, we flip a fair coin to determine random variable $b_i \sim \{0, 1\}$. If $b_i = 1$, we inject $X_{ref}^i$ and its corresponding fuzzy duplicates $\{X_j^i\}_{j=2}^{n_{dup}}$ into $D$. Conversely, if $b_i = 0$, neither $X_{ref}^i$ nor its fuzzy duplicates are included in the training dataset. Formally, the training dataset $D$ is defined as:

$$D = D_{orig} \cup \left( \bigcup_{i=1}^{C} \left( \{X_j^i\}_{j=1}^{n_{dup}} \cdot b_i \right) \right), \tag{M5}$$

where $D_{orig}$ represents the original dataset without any canary injections, and $\cdot$ denotes set inclusion conditioned on $b_i = 1$.

To quantify memorization, we apply MIAs[15,56,57] on $\mathcal{M}$, computing a membership score $\alpha(X_{ref}^i)$ for each $X_{ref}^i$ based on query outputs from $\mathcal{M}$. Each MIA methodology corresponds to a membership scoring function, $\alpha(X_{ref}^i)$. We select three methods to compute $\alpha(X_{ref}^i)$:

1. Loss attack from Yeom et al.[56], which uses the model loss computed on the reference canary. For the sequence of textual characters $X$, tokenized as $T(X) = \{t_1, \ldots, t_L\}$, we denote the loss of language model $\mathcal{M}$ with tokenizer $T$ as $\mathcal{L}_{\mathcal{M}}(X) = -\frac{1}{L}\sum_{i=1}^{L} \log(\mathcal{M}_\theta(t_i|t_1 \ldots, t_{i-1}))$. For the Loss attack, we thus consider: $\alpha = \mathcal{L}_{\mathcal{M}}(X)$.
2. Ratio attack from Carlini et al.[15], which uses the model loss divided by the loss computed using a reference model, or $\alpha = \mathcal{L}_{\mathcal{M}}(X)/\mathcal{L}_{\mathcal{M}_{ref}}(X)$. We use the same $\mathcal{M}_{ref}$ as used to generate reference canaries, i.e., Llama-2 7B[43].
3. Min-$K$% Prob from Shi et al.[57], which computes the mean log-likelihood of the $K$% tokens with minimum predicted probability in the sequence. More formally, $\alpha = \frac{1}{E}\sum_{t_i \in \text{Min-}K\%} \log(\mathcal{M}_\theta(t_i))$, where $E$ is the number of tokens in Min-$K$% and we consider $K = 20$.

The MIAs are evaluated over all $C$ reference canaries, representing a balanced set of member canaries ($b_i = 1$) and non-member canaries ($b_i = 0$). Specifically, membership scores are used to compute the area under the receiver operating characteristic curve (ROC AUC), which quantifies the attack's ability to distinguish between reference canaries included in and excluded from $D$. We denote the resulting ROC AUC of MIAs on $\mathcal{M}$ as $\widetilde{\phi}$.

### Experimental setup

We then measure how a target LLM memorizes across fuzzy duplicates by comparing MIA performances reached for fuzzy duplicates relative to the performance achieved for exact duplicates. The degree to which exact duplicates are memorized by a target LLM depends on the exact experimental setup $\mathcal{S}$, which we define by the following characteristics:

1. The set of reference canaries $X_{ref}^i$ used as members and non-members. Throughout this work, we consider synthetically generated sequences generated using the pretrained Llama-2 7B[43] as reference model $\mathcal{M}_{ref}$. Unless stated otherwise, we use temperature $\mathcal{T} = 1.0$.
2. The choice of pretrained model $\mathcal{M}_0$, which we continue pretraining on training dataset $D$ containing the fuzzy duplicates.
3. The hyperparameters used for continued pretraining. We keep this constant throughout this work, apart from the initial learning rate $\eta_0$. We opt to choose $\eta_0$ separately for each choice of pretrained model $\mathcal{M}_0$, as we find this to be a crucial hyperparameter when comparing training progress and the absolute level of memorization across model size and architecture.
4. The MIA methodology and performance metric used to measure memorization. Throughout this work, we use the ROC AUC computed on a balanced set of member and non-member reference canaries, and consider different MIAs from the literature defined by their scoring function $\alpha$.

We consider a range of models as the pretrained model $\mathcal{M}_0$. In most experiments, we consider GPT-Neo 1.3B[45] developed by EleutherAI. In further ablations, we also consider its smaller and larger versions, GPT-Neo 125M and 2.7B, and further consider three other models from different model families: Gemma-2B[46] developed by Google, Phi-2[47] developed by Microsoft and Llama-3.2-1B[48] developed by Meta. All models we consider are open-weights, and the pretrained model parameters are downloaded from Hugging Face.

In all of our experiments, we continue pretraining $\mathcal{M}_0$ on a dataset with documents $D$ containing reference canaries and their fuzzy duplicates, resulting in the target model $\mathcal{M}$. We inject canaries in a collection of books available in the public domain, i.e. the original dataset $D_{orig}$. We use an open-source library[95] to collect $\frac{C}{2} = 100$ books made available under a permissive license on Project Gutenberg[96]. We collect books added to Project Gutenberg after the release of GPT-Neo, which were thus likely not present in the original model's training dataset. The books we selected contain 8.8M tokens (GPT-Neo) in total. For each reference canary $X_{ref}^i$ for which randomly assigned membership bit $b_i = 1$, we inject the $n_{dup} = 10$ fuzzy duplicates at random into one book and use this collection of modified books as dataset $D$ for continued pretraining.

In all of our experiments, we apply continued pretraining on $\mathcal{M}_0$ using the modified books $D$ for 1 epoch. We use as maximum sequence length 2048 tokens, an effective batch size of 2 and an Adam optimizer with $\beta_1 = 0.9$ and $\beta_2 = 0.999$. We use a linear learning rate scheduler

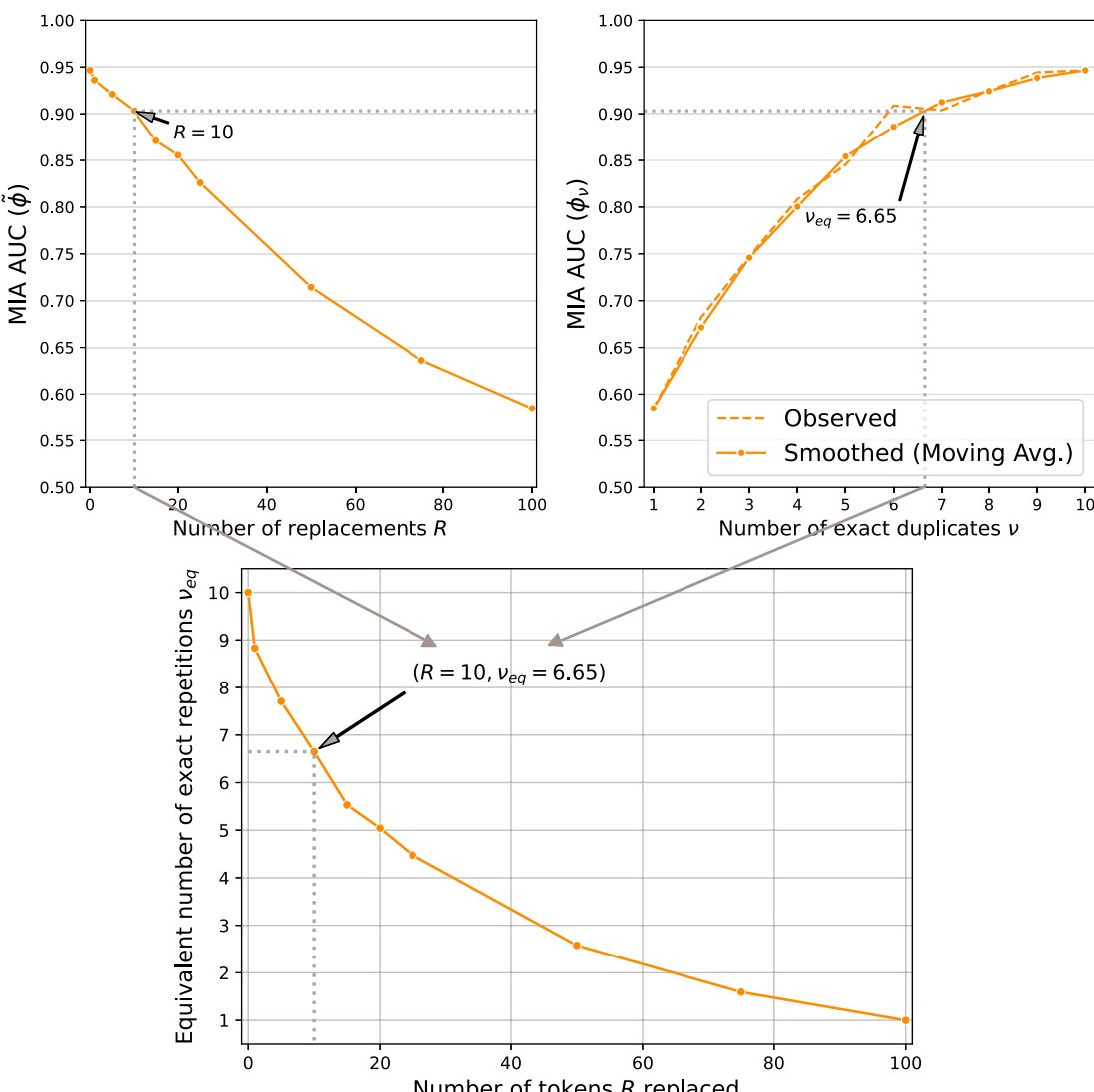

**Fig. 4 | Computing the equivalent number of exact duplicates $\nu_{eq}$ (results for one run for GPT-Neo 1.3B).** We first compute the MIA AUC $\tilde{\phi}$ for fuzzy duplicates with number of replacements $R$ (upper left), as well as the AUC $\phi_\nu$ for a range of exact duplicate counts $\nu$ (upper right). To avoid noise impacting the computation of $\nu_{eq}$, we smooth the evolution of MIA AUC with number of exact duplicates $\nu$ and provide both the observed and the smoothed values. Next, for each MIA AUC $\tilde{\phi}$ for a set of fuzzy duplicates, we interpolate to find the value of $\nu$ that achieves similar MIA performance in the same experiment setup, i.e., $\nu_{eq}$ is the $\nu$ for which $\tilde{\phi} \approx \phi_{\nu_{eq}}$.

with as initial learning rate $\eta_0$, final learning rate 0, and a weight decay 0.01. Across pretrained models $\mathcal{M}_0$, we empirically determine a value of $\eta_0$ to align with a training regime leading to a significant level of absolute memorization. We also show how the results vary with $\eta_0$ for GPT-Neo 1.3B (Fig. 5a).

For reference, continued pretraining for GPT-Neo 1.3B on $D$ takes roughly 2 GPU-hours on an NVIDIA A100 GPU, with minor variations for other models.

**Computing the exact duplicate equivalent**

We now detail the computation of the exact duplicate equivalent, $\rho$, for a given set of fuzzy duplicates generated by algorithm $\mathcal{A}$. First, we determine $\nu_{eq}$, which represents the equivalent number of exact repetitions that yield the same MIA AUC as observed for the fuzzy duplicates ($\tilde{\phi}$), under the same experimental setup $\mathcal{S}$. We then normalize $\nu_{eq}$ by the total number of fuzzy duplicates considered, ensuring that $\rho$ reflects the equivalent number of exact duplicates represented by a single fuzzy duplicate generated by $\mathcal{A}$.

To compute $\nu_{eq}$, we construct a modified training dataset $D_\nu$, where $\nu \in \{1, \ldots, n_{dup}\}$ exact copies of each reference canary $X_{ref}^i$ are injected. We use the same set of random variables $\{b_i\}$ generated during the fuzzy duplicate experiments. Formally, the modified dataset $D_\nu$ is defined as:

$$D_\nu = D_{orig} \cup \left( \bigcup_{i=1}^{C} \left( \{X_{ref}^i\}^\nu \cdot b_i \right) \right), \tag{M6}$$

where $\{X_{ref}^i\}^\nu$ denotes $\nu$ exact copies of $X_{ref}^i$. For each $\nu$, we train $\mathcal{M}_0$ on $D_\nu$ to get target model $\mathcal{M}_\nu$. We measure the performance of the MIA applied to $\mathcal{M}_\nu$ and denote this as $\phi_\nu$.

Figure 4 (top right) illustrates the observed values of $\phi_\nu$ for increasing values of $\nu$ for GPT-Neo 1.3B[45] as $\mathcal{M}_0$, initial learning rate $\eta_0 = 2e-5$, Ratio[15] as MIA methodology and reference canaries generated using temperature $\mathcal{T} = 1.0$.

We then leverage the values for $\phi_\nu$ to compute $\nu_{eq}$ for a given $\tilde{\phi}$. To prevent noise in the observed values from impacting the comparison of computed values of $\nu_{eq}$, we smooth the curve for exact duplicates

**Table 2 | Quantifying the mosaic memory across MIA methodologies**

| R | MIA methodology | | | | | |
| | Loss | | Ratio | | Min-$K$% Prob | |
| | AUC | $\rho$ | AUC | $\rho$ | AUC | $\rho$ |
|---|---|---|---|---|---|---|
| 0 | $0.942 \pm .006$ | 1 | $0.964 \pm .014$ | 1 | $0.968 \pm .006$ | 1 |
| 1 | $0.934 \pm .007$ | $0.935 \pm .014$ | $0.956 \pm .016$ | $0.898 \pm .031$ | $0.958 \pm .007$ | $0.906 \pm .030$ |
| 5 | $0.917 \pm .012$ | $0.801 \pm .058$ | $0.943 \pm .015$ | $0.781 \pm .023$ | $0.943 \pm .007$ | $0.786 \pm .051$ |
| 10 | $0.894 \pm .009$ | $0.663 \pm .030$ | $0.923 \pm .016$ | $0.646 \pm .021$ | $0.921 \pm .011$ | $0.657 \pm .029$ |
| 15 | $0.864 \pm .012$ | $0.536 \pm .029$ | $0.895 \pm .016$ | $0.520 \pm .036$ | $0.892 \pm .013$ | $0.533 \pm .017$ |
| 20 | $0.841 \pm .013$ | $0.455 \pm .027$ | $0.875 \pm .020$ | $0.452 \pm .038$ | $0.870 \pm .018$ | $0.455 \pm .024$ |
| 25 | $0.820 \pm .014$ | $0.399 \pm .014$ | $0.856 \pm .022$ | $0.401 \pm .018$ | $0.850 \pm .020$ | $0.401 \pm .011$ |
| 50 | $0.720 \pm .011$ | $0.180 \pm .013$ | $0.751 \pm .023$ | $0.186 \pm .012$ | $0.754 \pm .023$ | $0.196 \pm .015$ |
| 75 | $0.656 \pm .011$ | $0.076 \pm .010$ | $0.676 \pm .027$ | $0.081 \pm .013$ | $0.689 \pm .031$ | $0.095 \pm .012$ |
| 100 | $0.605 \pm .012$ | 0 | $0.612 \pm .020$ | 0 | $0.620 \pm .032$ | 0 |

MIA AUC and the exact duplicate equivalent $\rho$ for fuzzy duplicates (varying number of replacements R using $\mathcal{A}_{\text{replace}}$) across MIA methodologies. Mean and standard deviation across 5 independent runs for GPT-Neo 1.3B, initial learning rate $\eta_0 = 2e-5$ and reference canaries generated using temperature $\mathcal{T} = 1.0$.

using a moving average with window size of 3. Figure 4 (top right) shows how the smoothed curve compares to the observed values, confirming the changes to be minimal. Throughout this work, we leverage the smoothed curve for $\phi_\nu$.

For a given memorization level of fuzzy duplicates $\widetilde{\phi}$, $\nu_{\text{eq}}$ is determined as the value of $\nu$ for which $\widetilde{\phi} \approx \phi_{\nu_{\text{eq}}}$. We compute $\nu_{\text{eq}}$ through piecewise linear interpolation, or

$$\nu_{\text{eq}} = \nu' + \frac{\widetilde{\phi} - \phi_{\nu'}}{\phi_{\nu'+1} - \phi_{\nu'}}, \tag{M7}$$

where $\nu'$ represents the number of exact duplicates such that $\phi_{\nu'} \leq \widetilde{\phi} \leq \phi_{\nu'+1}$.

Figure 4 illustrates how for $R = 10$, we reach $\nu_{\text{eq}} = 6.36$. We then repeat the same process for each observation $(R, \widetilde{\phi})$, to quantify how $\nu_{\text{eq}}$ evolves with varying number of replacements $R$ across fuzzy duplicates. Note that, in rare cases, we empirically find $\widetilde{\phi}$ to reach values slightly larger than $\phi_{\nu = n_{\text{dup}}}$. We then also compute $\phi_\nu$ for values $\nu > n_{\text{dup}}$ until we find a $\nu_{\text{max}}$ for which $\widetilde{\phi} \leq \phi_{\nu_{\text{max}}}$ and only then apply the interpolation from Equation (M7).

Finally, we want a value reflecting the degree to which a fuzzy duplicate constructed by $\mathcal{A}$ contributes to the memorization of the reference canary, independent of the number of fuzzy duplicates ($n_{\text{dup}}$) we inject in the training dataset $D$. Therefore, we normalize $\nu_{\text{eq}}$ to get the exact duplicate equivalent, $\rho$:

$$\rho = \frac{\nu_{\text{eq}} - 1}{n_{\text{dup}} - 1}. \tag{M8}$$

Note that we need to subtract by 1 in both the numerator and denominator, as we always include 1 exact duplicate of the reference canary, or $X_1^i = X_{\text{ref}}^i$.

## Ablation studies

It is well known that the extent to which LLMs memorize depends on many factors of the experimental setup $\mathcal{S}$, including for instance training hyperparameters and model size[32]. We refer to this as the absolute level of memorization of the target model. For this reason, we have studied the mosaic memory throughout this work by computing the exact duplicate equivalent $\rho$, which captures the level of memorization relative to the absolute level reached for the specific experimental setup. To further study whether our quantification of the mosaic memory of LLMs holds for different levels of absolute memorization, we here vary the core elements of the experimental setup $\mathcal{S}$

and evaluate how $\rho$ varies with the number of replacements $R$ in $\mathcal{A}_{\text{replace}}$ across $\mathcal{S}$.

As a baseline, we use the results for GPT-Neo 1.3B[45] from Fig. 1(a). Specifically, we consider reference canaries generated with temperature $\mathcal{T} = 1.0$, and fuzzy duplicates generated using $\mathcal{A}_{\text{replace}}$ making "not evenly spread + not consistent" token replacements using the masked language model MLM and top $k = 10$.

First, we compute the value of $\rho$ versus number of replacements $R$ when we use a different MIA methodology. We consider the three MIA methods described above: Loss[56], Ratio[15] using the pretrained Llama-2 7B[43] as reference model and Min-$K$% Prob[57].

Table 2 reports the observed MIA AUC and the corresponding exact duplicate equivalent $\rho$ across replacements $R$ and MIA methodologies. We find $\rho$ to decrease similarly with increasing number of replacements $R$, independent of how the MIA AUC is computed or its absolute values. For instance, when $R = 20$ replacements are being made across fuzzy duplicates, we find an exact duplicate equivalent $\rho$ of 0.455, 0.452 and 0.455 for the attacks Loss, Ratio and Min-$K$% Prob, respectively.

This allows us to use $\rho$ as a metric to study the mosaic memory of LLMs (e.g., for various algorithms $\mathcal{A}$ to generate fuzzy duplicates), independent of the absolute level of memorization. As we find the Ratio MIA[15] to consistently lead to higher AUC values across setups, we consider this methodology further in this work.

Next, we study the mosaic memory across different values for the initial learning rate ($\eta_0$). As shown in Fig. 5a (left), the MIA AUC (measuring the absolute memorization) increases with $\eta_0$, achieving near-perfect performance (AUC of 1.0) for sequences with more than 6 exact duplicates when $\eta_0 = 3e-5$. This aligns with expectations: larger learning rates result in more substantial updates to model weights when a sequence is encountered during training, making the sequence's presence in the training data more detectable by the MIA.

We then continue to compute the exact duplicate equivalent $\rho$. Figure 5a (right) demonstrates that the progression of $\rho$ with respect to the number of replaced tokens ($R$) remains consistent across different values of $\eta_0$. This indicates that our relative memorization metric, $\rho$, is robust to variations in absolute memorization levels, enabling comparisons of LLM memorization across settings. Furthermore, this consistency suggests that LLMs exhibit a mosaic memory, i.e., effectively memorizing fuzzy duplicates, in a similar manner across varying hyperparameter choices, such as learning rate.

We repeat the same experiment while keeping the initial learning rate $\eta_0 = 2e-5$, yet now varying the size of the target LLM. Specifically, we also consider the smaller and larger GPT-Neo models[45], of 125 million and 2.7 billion parameters, respectively. Figure 5b (left) shows that

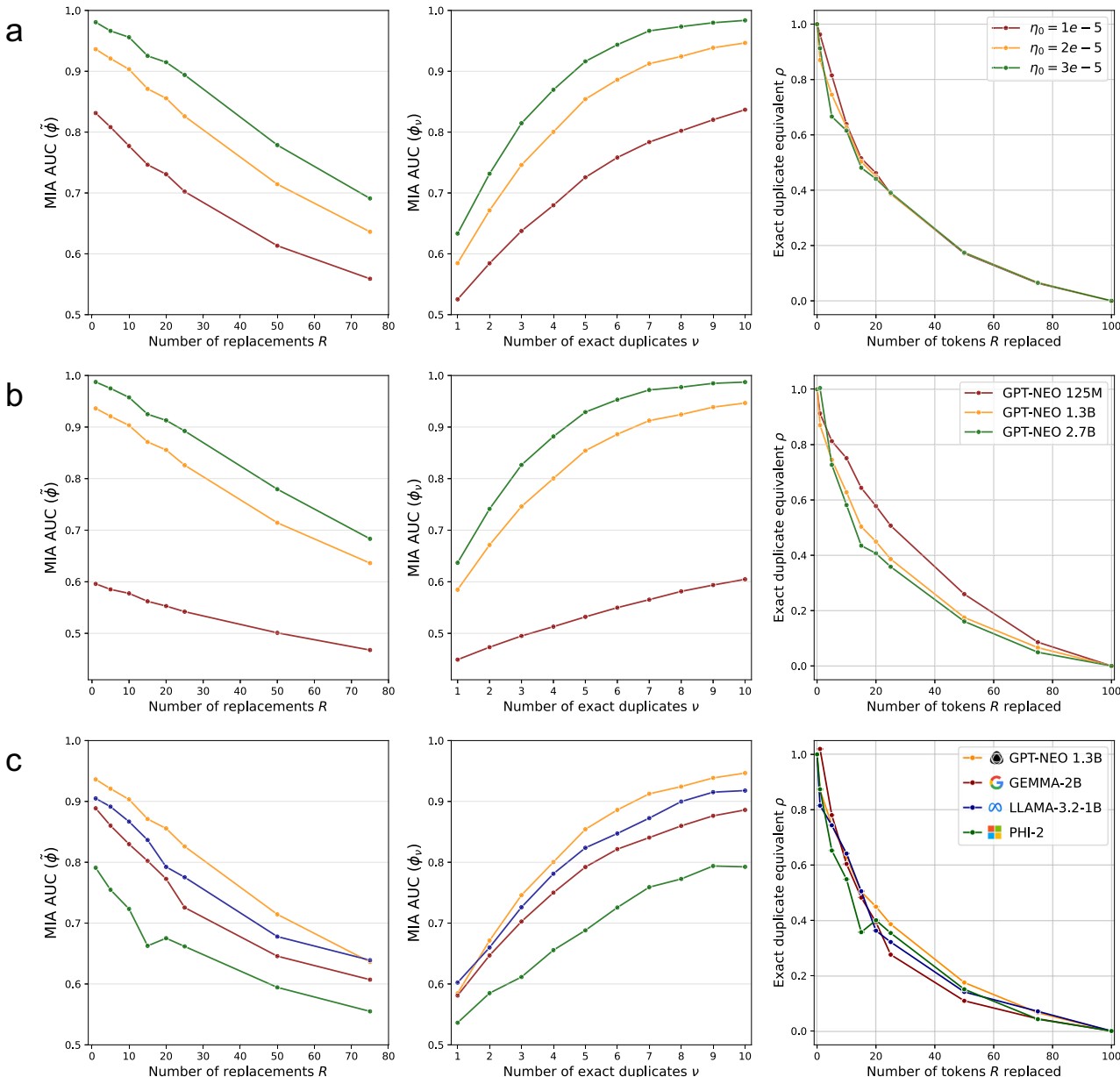

**Fig. 5 | Ablation studies for model training.** MIA AUC for increasing number of tokens replaced $R$ (left), MIA AUC for increasing number of exact duplicates $v$ (middle) and the corresponding exact duplicate equivalent $\rho$ for increasing $R$ (right). **a** Varying the initial learning rate $\eta_0$ for GPT-Neo 1.3B. **b** Varying the size of target model GPT-Neo for initial learning rate $\eta_0 = 2e-5$. **c** Varying the choice of pretrained model $\mathcal{M}_0$ using initial learning rates $\eta_0 = 2e-5$ (GPT-Neo 1.3B, Llama-3.2-1B), $\eta_0 = 3e-5$ (Gemma-2B) and $\eta_0 = 6e-5$ (Phi-2). Reference canaries are generated using temperature $\mathcal{T} = 1.0$.

the absolute level of memorization strongly depends on the model size. For instance, GPT-Neo 125M only reaches the MIA AUC of 0.61 at $v = 10$ exact duplicates, significantly lower than the near-perfect AUC reached for GPT-Neo 2.7B in the exact same setup. This again aligns with expectations: a model with more parameters has more capacity to memorize specific sequences, as also consistent with the literature[32].

We further examine the relative memorization of fuzzy duplicates ($\rho$). Our results indicate a similar mosaic memory across GPT-Neo 1.3B and GPT-Neo 2.7B models. Interestingly, GPT-Neo 125M exhibits slightly higher relative memorization, suggesting smaller models may rely more strongly on syntactic patterns, memorizing fuzzy duplicates proportionally more compared to their larger counterparts.

Next, we study the exact duplicate equivalent across model families. Figures 1a and 5c illustrate how $\rho$ evolves with the number of replacements $R$ across all pretrained models $\mathcal{M}_0$ considered in this

work[45–48]. Across model families, we empirically found a distinct impact of choice of initial learning rate $\eta_0$ and the absolute level of MIA performance and memorization. Across models, we empirically set the values for $\eta_0$ to have a meaningful level of absolute memorization (MIA AUC > 0.5): $\eta_0 = 2e-5$ for GPT-Neo {125M, 1.3B, 2.7B}[45], $\eta_0 = 3e-5$ for Gemma-2B[46], $\eta_0 = 6e-5$ for Phi-2[47], and $\eta_0 = 2e-5$ for Llama-3.2-1B[48]. Our results above show how the value of $\rho$ is fairly consistent across different values of $\eta_0$.

Thus far, we have only considered one set of reference canaries, namely the ones generated by sampling from the reference model $\mathcal{M}_{ref}$ (Llama-2 7B[43]) with temperature $\mathcal{T} = 1.0$. We note that $\mathcal{T} = 1.0$ is commonly used when generating synthetic text from pretrained LLMs, and we also empirically confirm that synthetic sequences generated using $\mathcal{T} = 1.0$ lead to meaningful sequences (see Supplementary Table 1).

We now repeat the baseline experiment for GPT-Neo 1.3B[45] as $\mathcal{M}_0$ and initial learning rate $\eta_0 = 2e-5$ while varying temperature $\mathcal{T}$. As such, we can verify whether our results are consistent regardless of the choice of reference canaries.

Figure 3a shows how $\rho$ decreases for increasing number of $R$ token replacements made across fuzzy duplicates, for different values of the temperature used to generate the reference canaries. We find that the memorization of fuzzy duplicates is strikingly consistent across a wide range of temperature values, even at $\mathcal{T} = 5$.

This is especially remarkable, as we empirically find sequences generated with $\mathcal{T} = 5$ to correspond to quite nonsensical sentences (see Supplementary Table 1). This suggests that LLMs memorize across fuzzy duplicates regardless of whether the reference canary is a coherent sequence of tokens, supporting the thesis that an LLM's mosaic memory is predominantly syntactic rather than semantic.

### Ablations for fuzzy duplicate generation techniques

In this section, we consider distinct ways of creating fuzzy duplicates not discussed in the main body of the paper.

The token replacement strategy $\mathcal{A}_{\text{replace}}$ discussed before inevitably leads to an information loss, as when tokens are replaced, the model sees fewer instances of the original tokens from the reference canary. We here investigate the impact of such information loss on memorization, separately from other properties of $\mathcal{A}_{\text{replace}}$. For this, we consider various fuzzy duplicate generation strategies, all of which maintain the same level of information loss, i.e., differing by $R$ tokens from the tokenized reference canary $T(X_{\text{ref}}^i)$:

1. Removing $R$ tokens at the end of the sequence (suffix). We here consider all fuzzy duplicates $\{X_j^i | j = 2, \ldots, n_{\text{dup}}\}$ to be equal to the first $|T(X_{\text{ref}}^i)| - R$ tokens of the reference canary.
2. Removing $R$ tokens at the start of the sequence (prefix). We here consider all fuzzy duplicates $\{X_j^i | j = 2, \ldots, n_{\text{dup}}\}$ to be equal to the last $|T(X_{\text{ref}}^i)| - R$ tokens of the reference canary.
3. Removing $R$ tokens randomly. We here remove the same $R$ random tokens for all fuzzy duplicates $\{X_j^i | j = 2, \ldots, n_{\text{dup}}\}$. The tokens to be removed are selected to be evenly distributed across the reference canary ("evenly spread + consistent"). This ensures that the subsequences between the replaced tokens are of equal length.
4. Replacing $R$ tokens. We here replace $R$ tokens by sampling another token from top $k = 10$ tokens predicted by the MLM. Again, the tokens to be removed are selected to maximize the length of the overlapping subsequences and are consistent across fuzzy duplicates ("evenly spread + consistent"). The random tokens used for replacement differ across fuzzy duplicates.

Figure 3b shows how the exact duplicate equivalent $\rho$ changes with the number of impacted tokens $R$ for the four kinds of information loss.

First, we find that removing the tokens from the prefix or suffix of the sequence maintains the most memorization for increasing $R$. This suggests that most information is retained by the model when the exact overlap between the reference canary and the fuzzy duplicates remains the largest, i.e., equal to $|T(X_{\text{ref}}^i)| - R$. Notably, removing tokens from the suffix leads to slightly higher values of $\rho$. We hypothesize this is because the MIA score is computed for the full length reference canary (i.e., $\alpha(X_{\text{ref}}^i)$). When tokens are removed only from the canary suffix instead of from its prefix, this means that the context used for token-level predictions is more consistent between model training and inference during the MIA, facilitating membership inference.

When the removed tokens are spread across the sequence, the memorization ($\rho$) drops quite sharply, yet remains significantly higher than the baseline memorization of $\rho = 0$. For instance, when $R = 20$ random tokens are removed (and the maximum overlapping subsequence between fuzzy duplicates and the reference canary is no

more than 4 tokens), the exact duplicate equivalent still remains as high as $\rho = 0.38$. This shows that the target LLM is able to memorize across missing segments of text. Finally, when we replace tokens with semantically meaningful ones instead of removing them, we observe a slight increase in $\rho$ for larger values of $R$. We hypothesize that token removal introduces additional noise compared to maintaining the coherence of the sequence by replacement, making it slightly harder for the LLM to memorize across fuzzy duplicates.

Previously, when replacing $R$ tokens, we distinguished between replacements evenly and not-evenly distributed within the reference canary and consistently and not-consistently across fuzzy duplicates. We now examine the corresponding impact on memorization.

In Fig. 3b, we replace $R$ tokens evenly distributed within the reference canary and replace the exact same tokens across fuzzy duplicates ("evenly spread + consistent"). We now also consider the cases in which the tokens to be replaced are not evenly distributed, both with the replaced tokens consistent across fuzzy duplicates ("not evenly spread + consistent") and not consistent ("not evenly spread + not consistent").

Figure 3c illustrates that the choice of token replacement positions has limited impact on the memorization of the fuzzy duplicates. Indeed, $\rho$ drops very similarly with increasing values of $R$ across all three scenarios. We do see a slight increase of $\rho$ when the tokens to be replaced are not uniformly spread across the reference canary. We hypothesize that the maximum overlapping subsequence between the reference canary and the fuzzy duplicate to be slightly larger in this case, which would help memorization. Notably, $\rho$ remains similarly high when the replaced tokens are not consistent across fuzzy duplicates, which shows how the target LLM memorizes across distinctly overlapping fragments of sequences.

### Impact of fuzzy duplicates on data extraction

To assess memorization and privacy risks in LLMs, prior work has examined not only vulnerability to MIAs but also training data extraction[15,18,32,33,58]. Carlini et al.[32] define a string $x$ as extractable from a model $\mathcal{M}$ if there exists a prefix $p$ such that the concatenation $[p\|x]$ appears in the training data of $\mathcal{M}$, and $\mathcal{M}$ reproduces $x$ exactly when prompted with $p$ using greedy decoding. Authors further use this definition to quantify memorization across model size, sequence length and repetition. Nasr et al.[58] later refer to this definition as discoverable memorization. Recognizing that exact string matching may be overly restrictive, Ippolito et al.[33] introduce the notion of approximate memorization, measured by the similarity between the generated text and the ground truth continuation $x$.

In this work, we study memorization through vulnerability to MIAs, motivated by two main considerations. First, defenses against MIAs are known to also offer protection against more severe effects such as training data extraction[78]. Note that training data extraction can be studied in various ways, ranging from generating text from a given prefix using greedy decoding with exact[15] or approximate[92] matching to more advanced methods considering stochastic decoding[97] or gradient-based optimization[98]. Second, also from a methodological perspective measuring memorization through MIAs is preferred, as text generation alone does not necessarily imply memorization but may instead reflect generalization[99].

We further investigate how the presence of fuzzy duplicates amplifies the risk of training data extraction. Following prior work[32,58], we provide the first $P$ tokens of a reference canary $T(X_{\text{ref}}^i)$ as prompt $p$ to the target model $\mathcal{M}$, i.e., $p = T(X_{\text{ref}}^i)[:P]$, and generate the next $|T(X_{\text{ref}}^i)| - P$ tokens via greedy decoding. In line with Ippolito et al.[33], we then measure approximate memorization by comparing the generated continuation to the true continuation $x = T(X_{\text{ref}}^i)[P:]$. We compute BLEU scores at the string level[33] and additionally report the normalized Levenshtein similarity on the token level, defined as one minus the Levenshtein distance divided by the continuation length.

**Table 3 | Quantifying the impact of fuzzy duplicates on the extraction risk of the reference canaries**

| $\mathcal{M}_0$ | R | Prefix length P | | | | |
| | | 50 | | 75 | | |
| | | BLEU | Levenshtein | BLEU | Levenshtein | |
|---|---|---|---|---|---|---|
| GPT-Neo 1.3B | 0 ($n_{dup}$ = 10) | 0.175 ± 0.245 | 0.209 ± 0.230 | 0.235 ± 0.341 | 0.317 ± 0.330 | |
| | 1 | 0.149 ± 0.209 | 0.184 ± 0.204 | 0.211 ± . 319 | 0.296 ± 0.313 | |
| | 5 | 0.134 ± 0.212 | 0.170 ± 0.208 | 0.178 ± 0.288 | 0.274 ± 0.292 | |
| | 10 | 0.108 ± 0.185 | 0.148 ± 0.176 | 0.160 ± 0.278 | 0.256 ± 0.284 | |
| | 15 | 0.101 ± 0.175 | 0.141 ± 0.171 | 0.144 ± 0.248 | 0.227 ± 0.257 | |
| | 20 | 0.096 ± 0.170 | 0.142 ± 0.175 | 0.133 ± 0.245 | 0.214 ± 0.250 | |
| | 25 | 0.080 ± 0.148 | 0.130 ± 0.152 | 0.107 ± 0.221 | 0.196 ± 0.247 | |
| | 50 | 0.062 ± 0.150 | 0.108 ± 0.150 | 0.086 ± 0.191 | 0.172 ± 0.215 | |
| | 75 | 0.058 ± 0.141 | 0.105 ± 0.147 | 0.075 ± 0.177 | 0.155 ± 0.193 | |
| | 100 ($n_{dup}$ = 1) | 0.062 ± 0.167 | 0.115 ± 0.167 | 0.077 ± 0.182 | 0.151 ± 0.215 | |
| GPT-Neo 2.7B | 0 ($n_{dup}$ = 10) | 0.308 ± 0.282 | 0.336 ± 0.273 | 0.405 ± 0.378 | 0.471 ± 0.354 | |
| | 1 | 0.276 ± 0.262 | 0.299 ± 0.262 | 0.337 ± 0.370 | 0.414 ± 0.355 | |
| | 5 | 0.237 ± 0.237 | 0.271 ± 0.238 | 0.252 ± 0.320 | 0.352 ± 0.321 | |
| | 10 | 0.170 ± 0.201 | 0.203 ± 0.195 | 0.203 ± 0.280 | 0.296 ± 0.286 | |
| | 15 | 0.142 ± 0.200 | 0.178 ± 0.187 | 0.191 ± 0.290 | 0.276 ± 0.290 | |
| | 20 | 0.111 ± 0.174 | 0.157 ± 0.181 | 0.157 ± 0.245 | 0.260 ± 0.267 | |
| | 25 | 0.106 ± 0.193 | 0.159 ± 0.185 | 0.153 ± 0.254 | 0.254 ± 0.251 | |
| | 50 | 0.073 ± 0.157 | 0.124 ± 0.155 | 0.111 ± 0.214 | 0.204 ± 0.222 | |
| | 75 | 0.069 ± 0.142 | 0.125 ± 0.149 | 0.086 ± 0.176 | 0.176 ± 0.214 | |
| | 100 ($n_{dup}$ = 1) | 0.059 ± 0.126 | 0.106 ± 0.130 | 0.095 ± 0.183 | 0.185 ± 0.223 | |

We prompt the target $\mathcal{M}$ on the first $P$ tokens from the reference canary and, in line with Ippolito et al.[33], measure the BLEU and normalized Levenshtein similarity between the continuation (greedy decoding) and the ground truth. We report the mean and standard deviation averaged across 100 reference canaries, for prompt lengths $P = 50$ and $P = 75$. As throughout this work, we consider GPT-Neo 1.3B and GPT-Neo 2.7B trained on a dataset with either $n_{dup}$ exact duplicates of the reference canaries or with fuzzy duplicates obtained by making $R$ replacements, using the setup from Fig. 1a for GPT-Neo 1.3B and the setup from Fig. 5b for GPT-Neo 2.7B.

Table 3 summarizes the extraction results for GPT-Neo 1.3B trained on a dataset that includes fuzzy duplicates of the reference canaries (using the same setup as in Fig. 1(a)). We evaluate two prompt lengths, $P = 50$ and $P = 75$, out of the total 100 tokens for each reference canary, and report the mean and standard deviation across all 100 reference canaries. We additionally consider the larger model, GPT-Neo 2.7B, which we expect to exhibit higher extractability.

We first consider approximate extraction in the presence of $n_{dup} = 10$ exact repetitions. For $P = 50$, we find a mean BLEU score of 0.18 for GPT-Neo 1.3B and 0.31 for GPT-Neo 2.7B. Exact extraction (BLEU of 1) occurs in 4 (GPT-Neo 1.3B) and 5 (GPT-Neo 2.7B) out of the 100 reference canaries. For only $n_{dup} = 1$ occurrence of the reference canaries, the mean BLEU score drops to 0.06 for both models.

Next, we find that fuzzy duplicates contribute substantially to the extractability of the reference canaries. Across models, we observe that as the number of replacements $R$ varies, both BLEU and Levenshtein similarity remain significantly above the baseline for $n_{dup} = 1$, and approach the levels observed for $n_{dup} = 10$ for lower values of $R$. Consistent with our findings on MIA vulnerability, these results demonstrate that fuzzy duplicates also contribute meaningfully to the extraction risk of reference canaries.

**Generating paraphrases as fuzzy duplicates**
In the results from Table 1, we use $\mathcal{A}_{paraphrase}$ to construct the fuzzy duplicates $\{X_j^i | j = 2, \ldots, n_{dup}\}$; i.e., we query an instruction-tuned LLM for $n_{dup} - 1$ paraphrases of the reference canary $X_{ref}^i$. Specifically, we query the models Meta-Llama-3-8B-Instruct[49], Mistral-7B-Instruct-v0.2[62] and GPT-4o[63] using the following system and user prompts:
1. System prompt: 'You are an assistant tasked with rephrasing the provided text in 9 different ways. Keep the original meaning intact (including the original natural or code language), but rephrase each version as if you are replacing the sentence entirely. Number

the rephrased sequences using 1. to 9. and separate each by '−', like this: '1. rephrase 1 − 2. rephrase 2 − .. − 9. rephrase 9'.'
2. User prompt: 'Can you rephrase the following sequence? $\{X_{ref}^i\}$'.

When using Meta-Llama-3-8B-Instruct[49] and Mistral-7B-Instruct-v0.2[62], we generate the answer from the model with a maximum number of new tokens of 1024 and temperature of 0.6 while sampling consecutively from the top predicted tokens for which the total predicted probability sums up to 0.9 (top-p) (as recommended in Meta-Llama-3-8B-Instruct[49] model card on Hugging Face). When using GPT-4o[63], we use the API's default sampling parameters, i.e., temperature and top-p equal to 1. We query the model until the processed output leads to $n_{dup} - 1$ distinct paraphrases. We then repeat the same process as in the main experiment, continuing pretraining the target model on a dataset always containing $X_{ref}^i$ and now its $n_{dup} - 1$ paraphrases. We provide example fuzzy duplicates obtained by paraphrasing for each instruction-tuned model in Supplementary Table 3.

Lastly, to put the results from Table 1 in perspective, we also compute the $n$-gram overlap between the fuzzy duplicates and the reference canary for fuzzy duplicates constructed by token replacements ($\mathcal{A}_{replace}$). Specifically, we also compute the mean and standard deviation overlap in $n$-grams (in GPT-Neo tokens) for the results from Fig. 1d for $k = |\mathcal{V}_{MLM}|$. Note that we consider all possible $n$-grams in a sequence and that an $n$-gram can appear in the fuzzy duplicate also when it is not in the same location. The results are summarized in Table 4.

**Collecting fuzzy duplicates from SlimPajama**
In this section, we provide further technical details on how we have collected fuzzy duplicates present in SlimPajama, while also providing additional results.

First, to define what constitutes a fuzzy duplicate, we consider a range of potential metrics to compute the distance between

**Table 4 | $n$-gram overlap for fuzzy duplicates constructed using $\mathcal{A}_{\text{replace}}$ with $k = |\mathcal{V}_{\text{MLM}}|$ (results from Fig. 1(d)), to put the results from Table 1 in perspective**

| Fuzzy duplicates | | $n$-gram overlap | | |
|---|---|---|---|---|
| using $\mathcal{A}_{\text{replace}}$ | $\rho$ | $n = 1$ | $n = 2$ | $n = 4$ |
| $R = 5$ | 0.73 ± 0.04 | 97.76 ± 1.20 | 91.49 ± 1.99 | 80.14 ± 2.95 |
| $R = 10$ | 0.54 ± 0.03 | 95.39 ± 1.82 | 84.08 ± 3.28 | 65.41 ± 4.66 |
| $R = 15$ | 0.43 ± 0.03 | 93.00 ± 2.44 | 76.87 ± 4.37 | 52.74 ± 5.66 |
| $R = 20$ | 0.34 ± 0.01 | 90.48 ± 2.96 | 69.84 ± 5.36 | 41.81 ± 6.24 |
| $R = 25$ | 0.29 ± 0.01 | 88.00 ± 3.50 | 63.09 ± 6.14 | 32.80 ± 6.53 |

Results (mean and standard deviation) aggregated over 5 independent runs for GPT-Neo 1.3B.

sequences. A sequence is then considered a fuzzy duplicate of a target sequence if both sequences are close according to this distance.

Throughout most of the experiments presented in this work, we have introduced a predefined perturbation ($\mathcal{A}$) to the reference canary and measured its impact on memorization by computing the corresponding exact duplicate equivalent $\rho$. Now, as we shift to analyzing a real-world dataset, we need a definition of fuzzy duplicates, i.e., a distance metric, that aligns with our prior experiments while also enabling us to estimate their impact on memorization in practical scenarios.

To this end, we consider a range of metrics to compute the distance between two sequences of tokens, and show how well each of them relate to the value of $\rho$. We consider the results of three of our main experiments, where we estimated $\rho$ values for token replacement ($\mathcal{A}_{\text{replace}}$, Fig. 1a), insertion ($\mathcal{A}_{\text{insert}}$, Fig. 1b) and shuffling ($\mathcal{A}_{\text{shuffle}}$, Fig. 1c), and compute the target distance between the reference canary and its fuzzy duplicates. We then replace experiment-specific distance metrics used before (number of replaced tokens $R$, number of inserted tokens $X_{\text{insert}}$ and Kendall tau distance $\tau$ respectively) with the target distance metric and plot the results of all three experiments on one graph with the target distance as a shared x-axis (Fig. 6a).

We consider the following distance metrics:

1. Levenshtein distance. This metric measures the minimum number of single-token operations (insertions, deletions, or substitutions) required to transform one sequence of tokens into another.
2. Levenshtein-Damerau distance. This metric extends Levenshtein distance by allowing transposition of adjacent tokens as a single operation, making it better suited for capturing typographical errors.
3. Longest Common Subsequence (LCS) distance. This metric calculates the length of the longest common subsequence (not necessarily consecutive) as the measure of similarity. The distance is then computed by subtracting the LCS length from the input length.
4. Token overlap (multiset). This metric calculates one minus the intersection of tokens between two sequences divided by the maximum number of tokens in either sequence. As such, it represents how many tokens appear in both sequences while accounting for token frequency.
5. Jaccard distance (token level). This metric represents the dissimilarity between two sequences by calculating one minus the ratio of the size of token intersection to the size of token union.
6. Jaccard distance (n-gram level). This metric measures the dissimilarity between two sequences by calculating one minus the ratio of the size of n-gram (in tokens) intersection to the size of n-gram union, capturing sequence-level differences.

Figure 6a shows that Levenshtein, Levenshtein-Damerau and LCS distances show good correlation between the distance and the $\rho$ value across all three experiments, effectively capturing fuzzy duplicates that contribute to mosaic memory. Token overlap (multiset) and token-level Jaccard distance both perform well for token

replacement and token insertions, but fail to capture shuffling—as both metrics are order-independent, they assign zero distance to all shuffled fuzzy duplicates. N-gram-level Jaccard distance ($n = 13$), on the other hand, suffers from the opposite problem: most sequence perturbations we have explored lead to very little overlap in 13-g, with the Jaccard distance values collapsing to near 1 for a wide range of $\rho$ values.

To identify fuzzy duplicates in a real-world dataset, we have chosen the Levenshtein distance. This metric shows equally good correlation as other well-performing metrics, is conceptually simple and naturally encapsulates types of sequence perturbations we explored previously (token replacement, insertion and shuffling).

We now estimate the prevalence of fuzzy duplicates as defined by our selected metric in a real-world dataset used for LLM training. As a dataset, we consider SlimPajama[50], which contains 627 billion GPT-Neo tokens. We then consider a sequence a fuzzy duplicate of a target sequence if their Levenshtein distance is below a certain threshold. We hypothesize that a substantial portion of the dataset would have a large number of such fuzzy duplicates, thus heavily contributing to model memorization.

First, we identify a set of target sequences for which we will seek fuzzy duplicates. These target sequences can be selected in various ways, and the exact selection criteria likely impacts the associated number of fuzzy duplicates. We hypothesize that the number of fuzzy duplicates identified for a certain target sequence has a positive correlation with the number of times this target sequence appears exactly in the dataset. Hence, we first group sequences by their number of exact repetitions in buckets to then search for fuzzy duplicates for target sequences sampled from each bucket. Note that we do not necessarily aim to show that all sequences in the dataset have fuzzy duplicates. Rather, we seek to show that a significant portion of the dataset has a large number of fuzzy duplicates, thus raising memorization concerns.

We adapt the code from Lee et al.[31] to build a suffix array and find all sequences of 100 tokens repeated more than once in the SlimPajama. We then consider sequences with a certain number of exact repetitions across the dataset: {100, 1000, 10000} (±1%). We estimate each of these buckets is large enough to raise concerns if many meaningful fuzzy duplicates are identified. Namely, the dataset has over 50 million sequences repeated at least 100 times, over 700,000 sequences repeated at least 1000 times, and over 30,000 sequences repeated at least 10000 times.

We then randomly pick 100 sequences from each bucket (100, 1000 and 10000 exact repetitions across the dataset) as our target sequences. To focus on meaningful memorization, we apply an additional selection criterion at this stage. Namely, we filter out sequences with too many repetitions of the same token, excluding the sequences where the number of unique tokens is below the 5th percentile across the dataset (46 unique tokens for 100-token sequence, see Fig. 6b). As such, we eliminate trivial sequences in our study, e.g. sequences where one token (often a space) is repeated many times, with only a few nontrivial tokens present.

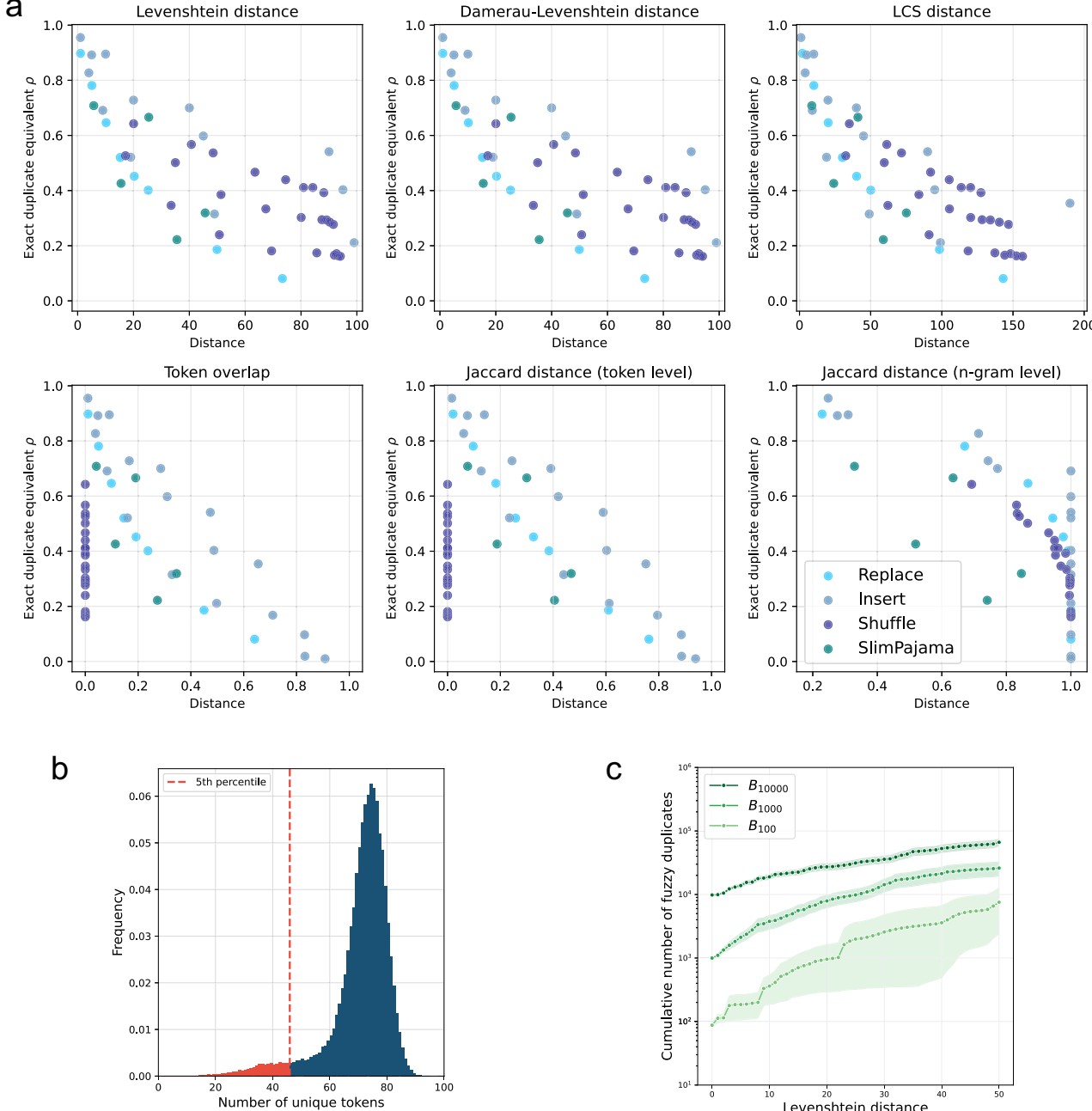

**Fig. 6 | Characterizing fuzzy duplicates. a** Relationship between the exact duplicate equivalent $\rho$ and mean distance between the reference canaries and their fuzzy duplicates across different distance metrics. We consider fuzzy duplicates obtained by token replacement $\mathcal{A}_{\text{replace}}$, insertion $\mathcal{A}_{\text{insert}}$ and shuffling $\mathcal{A}_{\text{shuffle}}$, as well as those found in a real-world dataset (SlimPajama) (results for GPT-Neo 1.3B). **b** Distribution of unique tokens across 100-token sequences in SlimPajama. The red line indicates the 5th percentile threshold (46 unique tokens) used to filter out trivial sequences with excessive token repetition. **c** Cumulative number of fuzzy duplicates at increasing Levenshtein distances for sequences with 100, 1000, and 10,000 exact duplicates in SlimPajama. Shaded regions represent the standard error of the mean.

Next, we scan the dataset with a moving window of 100 tokens and a 1-token-step, collecting all sequences with the Levenshtein distance to one of the target sequences below a certain threshold (in our case 50). As a computational optimization, we maintain the token overlap counter with the current window for each of the target sequences, allowing us to compute the relatively expensive Levenshtein distance in a rare event where the current window has at least 50 common tokens with the target sequence. We then avoid double counting by ensuring that each individual token is contributing to at most one fuzzy duplicate in the final count. To make the computation feasible on a very large dataset such as SlimPajama, we scan the first 5%

of a randomly shuffled dataset and extrapolate the results, assuming fuzzy duplicates are spread mostly uniformly across the shuffled dataset – which is confirmed by our observations on the positions of the exact duplicates. The scan took 96 h on 80 CPUs on a server with sufficient memory to fit the whole 5% portion of the dataset into memory.

In the main body of the paper (Fig. 2b), we provided the number of fuzzy duplicates for target sequences sampled from the bucket of sequences with 1000 ± 1% exact repetitions. Figure 6c now shows the cumulative number of fuzzy duplicates, averaged across sequences, for the other buckets as well, denoted as $B_{100}$, $B_{1000}$ and $B_{10000}$. Each

line starts at the respective number of exact duplicates (which is the selection criteria) and grows as we increase the Levenshtein distance. While the $B_{1000}$ bucket shows the fastest increase, all three buckets reach double the initial number of duplicates ($B_{100}$ by Levenshtein distance 8, $B_{1000}$ by 5 and $B_{10000}$ by 11). Moreover, for higher Levenshtein distances, $B_{100}$ and $B_{1000}$ reach a ten-fold increase compared to the initial number of exact duplicates (at Levenshtein distance 20 and 24 respectively).

In summary, these results show that even in datasets deduplicated on a document level, many exact duplicates still exist. For sequences repeated exactly, we furthermore recover a substantial amount of fuzzy duplicates at values of the Levenshtein distance which correspond to a meaningful contribution to memorization (see Fig. 2a).

### Example real-world fuzzy duplicates recovered from SlimPajama

We provide examples of real-world fuzzy duplicates identified in the SlimPajama dataset in Supplementary Table 4. We show both the target sequence (repeated exactly a certain number of times) and a selected subset of its fuzzy duplicates, along with their corresponding Levenshtein distance.

Interestingly, we find that fuzzy duplicates in real-world datasets can be remarkably subtle, e.g., minor formatting differences in software licenses or nearly verbatim excerpts from books. The presence of such fuzzy duplicates in a widely used dataset like SlimPajama, which has already undergone document-level deduplication, highlights that substantial content contributing to memorization can persist when deduplication is not performed at a finer granularity. Notably, all fuzzy duplicates shown here would remain even after sequence-level deduplication as commonly used by recent model developers to improve training efficiency[30,31,65,66] or mitigate privacy risks[30].

### Computing the exact duplicate equivalent for real fuzzy duplicates

Throughout this work, we have considered synthetic reference canaries and various algorithms for generating their fuzzy duplicates. We now extend this analysis to real-world fuzzy duplicates collected from SlimPajama, and compute their corresponding exact duplicate equivalent $\rho$.

We first examine the collected target sequences and their fuzzy duplicates, grouped by Levenshtein distance. Specifically, we define distance ranges [1, 10], [11, 20], [21, 30], [31, 40], and [41, 50] and, for each range, we identify target sequences for which at least $n_{dup} - 1$ unique fuzzy duplicates fall within the corresponding range.

Previously, we consider 100 target sequences for each of the three exact repetition levels (100, 1000, 10,000), giving a total of 300 potential reference canaries. After applying the requirement of having sufficient unique fuzzy duplicates per Levenshtein distance, we retain 50 reference canaries.

Following the same protocol as used throughout this work, we perform a membership inference attack using these real fuzzy duplicates. For each Levenshtein distance range, we inject half of the reference canaries and their fuzzy duplicates into training dataset $D$, while keeping the remaining half as non-members. We then continue pretraining one target model (GPT-Neo 1.3B) on $D$ per bucket. Since the reference canaries differ across distance, we recompute the MIA AUC $\phi_v$ for each number of exact repetitions $v$ and each Levenshtein distance range. Using these values, we then determine $\rho$ for each distance range.

We visualize these results in Fig. 2a. We observe that $\rho$ decreases with increasing Levenshtein distance, i.e., from $\rho = 0.708$ for distance [1, 10] to $\rho = 0.667$ for distance [21, 30] and $\rho = 0.319$ for distance [41, 50], consistent with the trends observed in our experiments using synthetic reference canaries and their fuzzy duplicates. This confirms that our findings generalize to real-world fuzzy duplicates, which

similarly contribute to the memorization of their corresponding reference canaries.

## Data availability

All data used in this paper is publicly accessible. To finetune all language models used as target models, we use books available in the public domain, i.e., the original dataset $D_{orig}$. We use an open-source library[95] to collect 100 books made available under a permissive license on Project Gutenberg[96]. We further generate reference canaries by sampling synthetic data from the open-source LLM Llama-2 7B[43]. Lastly, we investigate the presence of fuzzy duplicates in real datasets used for LLM training, namely in SlimPajama[50], which is publicly available on the open platform Hugging Face (https://huggingface.co/datasets/cerebras/SlimPajama-627B). Lastly, for all main graphs. Source data are provided with this paper.

## Code availability

The code necessary to reproduce the results in this work has been made publicly available in a Github repository (https://github.com/computationalprivacy/mosaic_memory[100]). All details are provided in the README.md file.

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

## Acknowledgements

We thank Marek Rei, Murray Shanahan, Pierre Colombo, and Manuel Faysse for useful discussions and feedback.

## Author contributions

I.S. and M.M. conducted the experiments and analyzed the results. I.S., M.M. and Y.-A. d.M. wrote the paper.

## Competing interests

The authors declare no competing interests.
