## [Transparent Peer Review file · Nature Communications]

The Mosaic Memory of Large Language Models

Corresponding Author: Professor Yves-Alexandre de Montjoye

Version 0:

Reviewer comments:

Reviewer #1

(Remarks to the Author)

This review has been performed in cooperation between junior and senior researchers, and we hence refer to ourselves in the plural "we".

We would like to thank the authors for the interesting work on a novel and timely topic that reads very well and is very accessible.

We will follow the outline of the nature review template throughout our comments.

First of all the paper asks an interesting research question and seems to provide interesting insights into the fact that LLMs, or foundation models, most likely do not memorize text not solely verbatim as exact token sequences, but also supported by additionally memorizing partial patterns of the original text. The authors also investigate if the models memorize "semantically" (or: in the latent space), or "syntactically" (as sequences of input tokens). Current ML research would suggest the latter, which is also supported by experiments of the authors.

We are not aware of prior work investigating these questions, and despite the fact that the second insight somewhat is to be expected, the first insight is interesting and innovative. It will have impact on the entire field of machine unlearning, as it implies that the attempt to remove representations of exact learned patterns will likely not suffice to guarantee privacy of the individuals corresponding to unlearned data.

We do believe that the results are likely correct, and that the study supports the conclusions and claims the authors are making. There are, however, a few minor things to consider, and probably revise, to make sure that the report is self-contained and the study can be replicated. None of the concerns is critical, all should be easy to address by revising the description of experiments and results.

First of all, the submission would benefit from further elaboration on the concept of mosaic memory. It is the main concept of the paper yet it is only briefly described in the introduction and could benefit from a more precise explanation: For instance, could you please clarify how the "sequences" and "overlapping fragments" (page 2) are defined in this context? The description given in the abstract remains rather vague. There is no further reference or discussion of the concept later-on in the text, and it remains a bit unclear which results exactly supported which claim regarding information being stored in "mosaic memory".

Second, we believe that making the paper more self-contained can help readers who lack necessary background knowledge. The article is submitted to a journal for all interested audiences, and it is unlikely that the audiences have deep knowledge of the field already. For example, the approach employed to generate the reference canaries is not explained, only cited. In this case, it would be valuable for the reader to know exactly how they are generated and the relation they have with the training data. Canaries that do not follow the same distribution of the training data could lead to biased results (it is well known that outliers are likely memorized), and thus it would be necessary to provide the correct justification that this does not happen. At the same time, if the canaries were chosen from the same distribution, there would need to be some verification that they have not already been part of the training data, as this would entirely invalidate any results from the experiments.

In addition, it would be necessary to explain how exactly the pretrained LLM M_0 is defined and how the further training on D is performed, as neither are explained. Without these explanations, we cannot be sure that there are no biases in the evaluation, and this knowledge is required for reproducibility. It also remains unclear if the original models would be required

- an important detail, as some indeed are not accessible, neither as open source nor in other ways to the scientific community in general.

Third, providing more intuitions on the experiments and metrics used in the evaluations can enhance the paper. For instance, please clarify how the fuzzy duplicates contribute to susceptibility to MIAs, and how that supports your claim of the memory being "mosaic". Regarding the metrics, there is no description of the ROC AUC or the "MIA performance", nor is it clear what they tell us exactly.

Also, the definition of ν_{eq} in "determine ν_{eq} as the value of ν for which $\tilde{\phi} \approx \phi_{eq}$ " is not clear. Is ν_{eq} defined as the ν that more closely approximates $\tilde{\phi}$? This is not clear in the text.

It remains unclear to us, over which models the fuzzy duplicates with token insertions and shuffling actually are tested? This is not specified in text or captions. The model employed could also be mentioned in the captions of the figures to improve readability.

Please clarify how the baselines of the fuzzy duplicates with token insertions are defined. On page 4, what does it mean that "n-grams are randomly scattered" and how is the insertion of infinite elements defined? The comparison to the baseline in the shuffling experiment is also not clear.

In the comparison to paraphrasing, the ρ in Table 1 is compared to that of Figure 3. However, as far as we understand, the figure and table are conducted under different models. Does this ensure a fair comparison?

Some minor suggestions and questions:

* While the experiment studying syntactic versus semantic memorization is interesting, is the conclusion that it is predominantly syntactic actually "surprising"? Isn't it well understood that machine learning actually learns and produces preexistent patterns, but does not understand/learn the semantic meaning of these patterns?

* In general, the section on "Quantifying a mosaic memory" is a bit hard to follow and could benefit from revision

* Please be sure to indicate in the introduction that you are working with 100 tokens. Otherwise, sentences such as "Levenshtein distance 10, for instance, represents just 10% of tokens" may confuse.

* Note that "baseline" is repeated in "(against the baseline baseline of $\rho=0.321$)".

* The labelling of Figure 5 would benefit from improvement. Could the actual n-gram sizes be included just next to the lines (for example, left to where they finish around $x=50$)? The color grading makes them very hard to distinguish. In addition, the in-text references to specific curves may not be immediately clear, especially since the first line from the bottom is almost invisible except for the slight increase near the end. It is easy then to miss that curve when finding the "fifth line from the bottom", which the text highlights. In addition, wouldn't the curve for $n=25$ mentioned on page 12 be actually the one for $n=20$? It seems that the curves are in 10-step intervals, but it is hard to confirm due to the similar color tones.

* Please note that there are minor misprints in some references, such as 49 and 61.

(Remarks on code availability)

Reviewer #2

(Remarks to the Author)

(Remarks on code availability)

Reviewer #4

(Remarks to the Author)

Key results

The motivation of the study is to reduce sensitive information leakage of LLMs due to fuzzy duplications in the training dataset. The "sensitive information" is simulated by synthetic sequence generated using an LLM, which the authors refer as "canaries". "Memorization" is measured using Membership Inference Attacks (MIAs). Specifically, canaries along with their 'fuzzy' duplicates (constructed by replacement, insertion, paraphrases, etc.) are used for additional training a pre-trained LLM. Their (log) probabilities after training are compared to canaries that are not used in training. The main result of the paper is measuring the contribution of the presence of fuzzy duplicates to the memorization of the canaries. For this they introduce a relative measure of the memorization, exact duplicate equivalent (ρ), which evaluate how many fuzzy duplications are equivalent to one exact duplication. The authors found that, depending on the degree of fuzziness (e.g. the

number of replacements, Levenshtein distance), a small number of fuzzy repetitions can have a similar effect (e.g. sequence probability) as having one exact repetition. In the later parts of the study, the authors examine a popular deduplicated dataset and show that it still contains many fuzzy duplicated sequences. Using a more aggressive deduplication procedure, they can further remove fuzzy duplications in that dataset.

Validity

The experiments are mostly based on the existing methods, such as Membership Inference Attacks, Meeus, M (2024). The methods and results are valid. However, the authors may overinterpret the results out of the scope of the experiments. The actual implication of the results is vague and complicated. There is almost no statistical test on any results.

The main measure, ρ , is used to make most of the claims in the manuscript. However, ρ measures the effect of fuzzy duplicates relative to exact duplicates. Though not claimed directly, the author imply that higher ρ is undesirable, because higher ρ means the fuzzy duplicates are as “detrimental” as exact duplicate. However, it is not clear how bad the exact duplicates are. Thus, it is difficult to relate ρ to impact on realistic use cases. For example, ρ does not tell how many (fuzzy) duplicates are sufficient to induce unsafe information leakage and with what probability. It is also unclear whether removing duplication can eliminates information leakage, given that “personal” information which is a main topic of interest in this context, is typically not widely known (repeated in the training dataset).

Based on the results in Fig. 1, the authors claim that “fuzzy duplicates contribute significantly to memorization.” This is a questionable claim. The claim comes from the observation that on all “number of tokens R replaced” except 100 the ρ is larger than 0. This observation could be trivial. Two points on the plot ((0, 1.0) and (100, 0)) are there by construction. Simply connecting the two points result in all middle values larger than 0. More quantitative claim “In practice this means that having two fuzzy duplicates in the training data with 10% of the original tokens replaced yields higher memorization in the than one exact repetition” is technically true, but it is not clear what the memorization effect of one exact repetition is.

Another issue is that the main experiments are about finetuning pretrained LLMs (A minor terminology suggest: the authors describe their training procedure as “further pretraining” of a pretrained model. However, “Continued pretraining” is a more prevalent name than “further pretraining” by convention.).

However, the role of different finetuning protocols is not discussed. For example, the learning rate could affect ρ . More importantly, it is unclear whether memorization of sequences introduced in this fine-tuning is similar to memorization of sequences introduced in the complex and long pre-training process.

Significance

Understanding the memory of LLMs is a fundamentally important topic. A main claim of the manuscript is that the fuzzy duplicates in the training data can lead to undesired memorization. Additional interesting result is the observation that syntactic more than semantic, similarity determines the efficacy of fuzzy duplicates. However,

(1) The fact that memorization can be enhanced by inexact repetition is not surprising, and is clearly demonstrated by recent work on paraphrases. The general significance of the quantitative measure adopted here is questionable, as explained above. (2) In the context of data privacy and copyright protection, memorization is undesired. However, in a broader sense, memorization is often thought to be a desired property of an LLM. Remembering more factual details can reduce model hallucination. The manuscript does not explain the tradeoff of the two. The “canaries” which are used as a probe are neutral sequences (i.e. general natural language content generated by an LLM with $T=1$). It is unclear whether they are a good model of desired or undesired memorized items.

There are well-recognized studies showing paraphrases are necessary for generalization (for example Zeyuan Allen-Zhu 2024 part 3.1). The authors do not discuss how they fuzzy duplicates and paraphrases help generalization but only discussed their role on “memorization”.

Data and methodology

There is no issue on the data. The methods are clearly explained in the supplementary. The LLM models used in the experiments are two generations behind, but still acceptable.

Analytical approach

There is no statistical analysis except standard deviation in the tables. Adding statistical analysis to most of the plots will be helpful.

Suggested improvements

The authors state that the deduplication process of SlimPajama fails to even remove exact duplicates. However, the authors did not discuss potential reasons of the failure. It is surprising that a widely-used dataset fails so evidently. Some examples would be helpful to understand the failure. One potential reason I can think of is that sequence length used in the SlimPajama deduplication and in this manuscript are different, so that there are shorter sequences that are exactly the same. As the authors state in the section of “deduplication”, there is a trade-off in the the choice of deduplication approaches. However, the authors only show the effectiveness of sample-level deduplication (Fig. 5). The result in Fig. 5 is as expected but does not answer important questions like what the optimal deduplication procedure is.

Clarity and context

The manuscript is clearly written.

References

As noted in the “Significance”, references on the general role of paraphrases and generalization should be included.

(Remarks on code availability)

N/A

Reviewer #5

(Remarks to the Author)

(Remarks on code availability)

The code will be made publicly available, but not yet.

Reviewer #6

(Remarks to the Author)

There has been a long line of work analyzing the memorization of training examples in the parameters of neural networks, which in recent years has been extended to large language models (LLMs). In LLMs, such memorization typically occurs due to duplicates in the training data, i.e., the same document or string being present multiple times. While previous work has predominantly looked at memorization resulting from exact duplicates, the present paper shows that memorization happens even if the duplicates are fuzzy, for example if some of the tokens in them are swapped or the strings are rephrased versions of the same string. By carefully choosing the fuzzy duplicates to be at varying semantic distances from the original string, the authors show that memorization is rather syntactic than semantic: operations that increase semantic distance have a smaller effect on memorization than those that increase syntactic distance. For measuring memorization, the authors use the ROC AUC of different membership inference attacks (MIAs) from the literature and introduce the metric ρ to relate the memorization of fuzzy duplicates to the memorization of exact duplicates.

Given the potential impacts of the memorization of training data (leakage of confidential information, copyright implications, etc.), mitigation of memorization and of membership inference and extraction attacks is an important problem. The authors show that mitigation techniques relying on exact deduplication do not suffice. This is an important contribution with potential real-world impacts, and is worth being published. On the more foundational side, the findings that point towards models remembering syntactically rather than semantically are an interesting contribution to ongoing discussions about the internal workings of LLMs and their abstraction abilities.

On the methodological side, the paper is sound. The authors perform many ablation studies, which are particularly crucial for such synthetic experiments. These ablations largely show the robustness of the findings. The main methodological contribution of the paper is the duplicate equivalent metric ρ , which measures how many exact duplicates a fuzzy duplicate corresponds to. This is an intuitive metric because it is mostly independent of the absolute level of memorization, and seems useful for future work. The other methodological contributions are all quite natural, but not particularly innovative.

One methodological criticism that I have: generating reference canaries via sampling from an LLM could lead to generating sequences that are memorized by that LLM, which are in turn likely to occur multiple times in the training data because they are memorized. The authors use Llama 2 as the reference model, which they do not use for evaluating memorization, but it is still likely that at least some of its training data overlaps with that of the models that they evaluate. Following this logic, it might be that the canaries are sequences that are already duplicated in the models' training data, thus distorting the ρ values. This criticism is partly addressed through ablation studies with higher sampling temperatures for the reference canaries.

There are two additional types of experiments that would make the paper stronger.

The first is an experiment that brings it all together. The authors show that memorization occurs with synthetic canaries and fuzzy duplicates, and show that such fuzzy duplicates occur in the LLM training dataset SlimPajama. A very natural next step, and the final proof that memorization due to fuzzy duplicates is a real problem, would be training an LLM on (a subset of) a dataset such as SlimPajama and measuring the memorization of strings with naturally occurring fuzzy duplicates. Even though the algorithms used for generating the canaries and for producing fuzzy duplicates used by the authors are reasonable, the resulting sequences still remain synthetic, and real-world text and fuzzy duplicates will look different. The authors already detect fuzzy duplicates and estimate their frequency in SlimPajama, so running their training and MIA attacks with this data doesn't seem like a large amount of effort. This is just my outside perspective, though, and I might be overlooking challenges with such an experiment.

Many of the risks of memorization only materialize if it is not only possible to determine that a string was used during training, but if (parts of) it can be extracted from the model. Carlini et al. define the concept of extractable memorization [1,2]: A string s is extractable from a model if one can construct a prompt so that the model generates s . See also Lukas et al. for the case of PII [3], and the work connecting membership and attribute inference [4,5]. In addition to MIA attacks, the authors could have also tested extraction attacks to see whether those also still work in the presence of only fuzzy duplicates. In the absence of that, it would still be good if they could (briefly) comment on the difference between membership inference and extraction.

The first of those two proposed (series of) experiments is more important to me than the second, which can rather be seen as a suggestion for future work. In case the first experiment would be significantly more challenging and a larger amount of work than I am thinking right now, I would suggest accepting the paper even without it.

Lastly, I have a few questions for the authors:

1. For the token replacement via RoBERTa, how do you deal with the fact that this model has a different tokenizer than some of the models that you evaluate memorization on? E.g., if you want to replace one token, but the token returned by RoBERTa is actually tokenized into two tokens by the tokenizer of the model that you evaluate? This would be good to clarify in the

paper.

2. "the respective ρ also rise from 1.99 to 2.51 and 3.72, respectively" (p.8, top). Those seem to be the wrong numbers, as ρ should be between 0 and 1. Do you mean the numbers from Table 1, i.e., 0.11, 0.17, 0.30?

3. In M8 you write that you did not control the sampling parameters for GPT-4o. Was it not possible to set the temperature back when you used the API for these experiments, or does the statement only refer to the probability sum?

[1] Carlini et al. Extracting Training Data from Large Language Models. 2023. USENIX

[2] Nasr et al. Scalable Extraction of Training Data from (Production) Language Models. 2023. arXiv

[3] Lukas et al. Analyzing Leakage of Personally Identifiable Information in Language Models. 2023. SP

[4] Yeom et al. Privacy Risk in Machine Learning: Analyzing the Connection to Overfitting. 2018. CSF

[5] Salem et al. SoK: Let the Privacy Games Begin! A Unified Treatment of Data Inference Privacy in Machine Learning. 2023. SP

(Remarks on code availability)

Version 1:

Reviewer comments:

Reviewer #1

(Remarks to the Author)

We would like to thank the authors for their hard work in improving the manuscript. We believe that all of the comments and questions raised in our review have been addressed in this new version or in the author's response.

Overall, we are grateful for the effort made to make the paper more accessible and for all the additional descriptions and specifications provided. We believe these improvements have substantially enhanced the paper. Nevertheless, we have a few final minor comments about this new version.

Firstly, we appreciate the detailed answers given in the Author's Response, but we believe that some have not been fully addressed in the final version of the manuscript itself. For example, the fact that " ρ captures how fuzzy duplicates influence memorization regardless of how inherently "memorable" a sequence is", or that ν_{eq} is "the value of ν for which $\tilde{\phi}_{\nu}$ best approximates $\tilde{\phi}$ " are very clear in the Author's response, but still not as clear in the newer draft (in particular, note that the latter is much clearer in the author's response than in the new version "the value of ν for which $\tilde{\phi} \approx \phi_{\nu_{eq}}$ ", which is somewhat ambiguous). Ideally, we would like the authors to incorporate the specifications from the Author's Response into the text.

Similarly, regarding our comment on the "surprisingness" of the results, we appreciate the authors' explanation and acknowledge that the issue is more contentious than we previously thought. We believe that the answer provided in the authors' response is important, and we would like it to be mentioned or discussed in the paper as well. This would slightly increase transparency with regards to the current knowledge of the community.

We also think that the additional references to the appendices will help readers, as we can now see that they address some of our initial questions. However, we noticed that the rewriting has introduced a few terms into the main body that might require explaining. For instance, the term "temperature" in the added text "setting the temperature $T=1$ to mimic realistic text" is not explained in the main body.

We also realise that our comment on the canaries chosen by the same distribution was incorrect. We appreciate the authors helping us to better understand, and we agree with their response to our comment.

Finally, we noticed some typos:

* The T in temperature (pg 3) should be in \mathcal{T} in concordance with the appendices and to avoid confusion with the tokenizer T .

* Missing spacing after "indicating stronger memorization" (pg 3).

* Missing words in "yields higher memorization in the than one exact repetition" (pg 4).

We would like to thank the authors once again for their responses and corrections.

(Remarks on code availability)

Reviewer #2

(Remarks to the Author)

I co-reviewed this manuscript with one of the reviewers who provided the listed reports. This is part of the Nature Communications initiative to facilitate training in peer review and to provide appropriate recognition for Early Career

Researchers who co-review manuscripts.

(Remarks on code availability)

Reviewer #4

(Remarks to the Author)

I would like to thank the authors for their detailed response to my comments. I appreciate the effort taken to address the concerns raised regarding validity, significance, and methodology.

Statistical Robustness: I appreciate the inclusion of the mean and standard deviation across independent runs for the main results.

Clarification of Scope: The authors have provided a clear distinction that their work aims to study the mechanism of memorization rather than making normative judgments of the consequences of memorization. Though I still believe the pure empirical approach could only give limited insights to the mechanism of memorization, and interesting and significant problems, such as the trade-off between data deduplication and generalization, are left to future works.

Comparison to Prior Work: I accept the distinction drawn between this study and the work of Allen-Zhu et al.

SlimPajama Deduplication: The clarification regarding the difference between document-level (Jaccard similarity) and sequence-level (Levenshtein) deduplication effectively explains why fuzzy duplicates persist in the dataset.

Overall, the revisions and additional explanations have strengthened the manuscript. I am mostly satisfied with the authors' response.

(Remarks on code availability)

Reviewer #5

(Remarks to the Author)

(Remarks on code availability)

Reviewer #6

(Remarks to the Author)

Creation of the canaries: There exist examples in the literature where models prompted with just the beginning-of-sequence token have produced sequences from the training data, e.g., [1, 2]. I am, however, sufficiently convinced by the experiments with higher sampling temperatures.

While canaries that are already duplicates could influence the concrete value of ρ , this should not affect the trends visible in the experiments and the main conclusions. Overall, I am not too concerned about this potential issue.

Experiment with real-world fuzzy duplicates: It's great that the authors have conducted the proposed experiment. It adds a lot to the authors' argument that fuzzy duplicates should be taken seriously. The results are less clear than in the controlled experiments, but still show a trend of ρ decreasing with increasing distance between the fuzzy duplicates. In future work, it would be interesting to analyze the real-world fuzzy duplicates more closely to, e.g., identify properties of the duplicates within each bucket that might explain the unexpected jumps in ρ .

Experiment measuring extraction risk: Thank you for conducting this additional experiment. The findings are convincing and show the robustness of the impact of fuzzy duplicates on memorization and their risks beyond membership inference.

The questions that I had for the authors have been fully addressed.

I am very satisfied with the authors' response and with the changes they made to the paper and recommend acceptance.

[1] Lee et al. Deduplicating Training Data Makes Language Models Better. 2022. ACL.

[2] Carlini et al. Extracting Training Data from Large Language Models. 2021. USENIX

(Remarks on code availability)

The Mosaic Memory of Large Language Models (NCOMMS-25-29287): Response to reviewers

We thank the reviewers for their valuable feedback and the time they dedicated to our manuscript. In this document, we carefully respond to each reviewer's comments. Specifically, for each reviewer, we reproduce their comments verbatim and then provide our responses in blue. Our responses either explain how we have addressed the feedback in the revised manuscript or offer additional clarifications where appropriate.

Reviewer #1 (Remarks to the Author and **Authors' Response**)

This review has been performed in cooperation between junior and senior researchers, and we hence refer to ourselves in the plural "we".

We would like to thank the authors for the interesting work on a novel and timely topic that reads very well and is very accessible.

We thank the reviewers for their time spent on the manuscript and for their constructive and thoughtful feedback, which helped us improve the manuscript.

We will follow the outline of the nature review template throughout our comments.

First of all the paper asks an interesting research question and seems to provide interesting insights into the fact that LLMs, or foundation models, most likely do not memorize text not solely verbatim as exact token sequences, but also supported by additionally memorizing partial patterns of the original text. The authors also investigate if the models memorize "semantically" (or: in the latent space), or "syntactically" (as sequences of input tokens). Current ML research would suggest the latter, which is also supported by experiments of the authors.

We are not aware of prior work investigating these questions, and despite the fact that the second insight somewhat is to be expected, the first insight is interesting and innovative. It will have impact on the entire field of machine unlearning, as it implies that the attempt to remove representations of exact learned patterns will likely not suffice to guarantee privacy of the individuals corresponding to unlearned data.

We agree and have now added a paragraph in the Discussion section highlighting the implications of our findings for the field of machine unlearning.

We do believe that the results are likely correct, and that the study supports the conclusions and claims the authors are making. There are, however, a few minor things to consider, and probably revise, to make sure that the report is self-contained and the study can be replicated. None of the concerns is critical, all should be easy to address by revising the description of experiments and results.

First of all, the submission would benefit from **further elaboration on the concept of mosaic memory**. It is the main concept of the paper yet it is only briefly described in the introduction and could benefit from a more precise explanation: For instance, could you please clarify how the "sequences" and "overlapping fragments" (page 2) are defined in this context? The description given in the abstract remains rather vague. There is no further reference or discussion of the concept later-on in the text, and it remains a bit unclear which results exactly supported which claim regarding information being stored in "mosaic memory".

We thank the reviewer for highlighting this point. We agree that the concept of *mosaic memory* – one of the key concepts in our work – was not defined sufficiently in the original version. We have therefore expanded its description in the Introduction (second paragraph). Specifically, we define mosaic memory as the model's ability to memorize an arbitrary sequence of tokens through exposure to its fuzzy duplicates -- fragments of text similar to the original sequence with some tokens missing, replaced, or shuffled. Mosaic memory manifests in the increased memorization from observing fuzzy duplicates during training – similarly to how exact repetitions amplify it.

Second, we believe that **making the paper more self-contained can help readers who lack necessary background knowledge**. The article is submitted to a journal for all interested audiences, and it is unlikely that the audiences have deep knowledge of the field already. For example, **the approach employed to generate the reference canaries is not explained**, only cited. In this case, it would be valuable for the reader to know **exactly how they are generated and the relation they have with the training data**.

We thank the reviewer for this helpful suggestion. We agree that making the paper more self-contained will benefit readers who may not have detailed background knowledge of the field. We have therefore added clarifications in the main text (section "A metric to study the memorization of fuzzy duplicates"), and now more explicitly refer to the detailed experimental setup in sections M1-6.

Canaries that do not follow the same distribution of the training data could lead to biased results (it is well known that outliers are likely memorized), and thus it would be necessary to provide the correct justification that this does not happen.

We agree that the canaries are not drawn from the same distribution as the training data, as they are synthetically generated rather than written by human authors. However, we believe this does not affect the validity of our findings. Our analysis focuses on the *relative* memorization of fuzzy duplicates, measured using the exact duplicate equivalent (ρ), which is designed to be invariant to the *absolute* memorization level of individual sequences. In other words, ρ captures how fuzzy duplicates influence memorization regardless of how inherently "memorable" a sequence is.

Our main experiments use canaries generated with temperature 1, producing text sampled from the base model's learned distribution and not intended to be outliers. In Section M6, we further vary the generation temperature between 1 and 5, where higher temperatures yield rarer, more unexpected sequences that are indeed more prone to memorization. The results

remain consistent across this range, confirming that ρ is robust to changes in the absolute memorization level.

At the same time, if the canaries were chosen from the same distribution, there would need to be some **verification that they have not already been part of the training data**, as this would entirely invalidate any results from the experiments.

We thank the reviewer for raising this subtle but important point. While it is unlikely that any of our synthetically generated 100-token canaries (sampled with temperature 1) appear verbatim in the pretraining data, we agree that, given the immense scale of modern datasets, it is difficult to rule out that some exact or fuzzy duplicates might exist.

We respectfully disagree, however, that such overlap would invalidate our findings. Our experiments were specifically designed to remain valid even in this case.

First, we perform the MIA specifically against the continued pretraining stage, where we explicitly control the injection of canaries. Both member and non-member canaries are generated using the same process and are therefore equally likely to have (fuzzy) duplicates in the pretraining corpus. Any overlap would thus affect both groups equally and not invalidate the MIA.

Second, while such overlap could in principle affect the absolute memorization level, our analysis focuses on relative memorization – quantified through ρ – which measures how fuzzy duplicates compare to exact duplicates under the same setup. As shown in Section M6, our results are consistent across a range of hyperparameters that influence the overall level of memorization, confirming that ρ is robust to such variations. This includes the case in which canaries are generated with a temperature up to 5, where overlap with pretraining data becomes exceedingly unlikely.

In addition, it would be necessary to **explain how exactly the pretrained LLM M_0 is defined and how the further training on D is performed**, as neither are explained. Without these explanations, we cannot be sure that there are no biases in the evaluation, and this knowledge is required for reproducibility. It also remains unclear if the original models would be required - an important detail, as some indeed are not accessible, neither as open source nor in other ways to the scientific community in general.

We thank the reviewer for pointing this out and agree that additional details would improve clarity and reproducibility. We have now expanded Section M4, where we provide a detailed description of the data collection and model training procedure, along with all relevant hyperparameters. We also now explicitly refer to this in the main text.

For the model availability, we only consider open-weights models, available on Huggingface (GPT-Neo (EleutherAI), Gemma-2B (Google), Phi-2 (Microsoft), and Llama-3.2-1B (Meta)).

We furthermore ensured that all our experiments are fully reproducible using the open-source code we provide alongside the paper, and only rely on publicly accessible data and pretrained models.

Third, **providing more intuitions on the experiments and metrics** used in the evaluations can enhance the paper. For instance, please clarify how the fuzzy duplicates contribute to susceptibility to MIAs, and how that supports your claim of the memory being "mosaic". Regarding the metrics, there is no description of the ROC AUC or the "MIA performance", nor is it clear what they tell us exactly. Also, the definition of ν_{eq} in "determine ν_{eq} as the value of ν for which $\tilde{\phi} \approx \phi_{eq}$ " is not clear. Is ν_{eq} defined as the ν that more closely approximates $\tilde{\phi}$? This is not clear in the text.

We thank the reviewer for highlighting the parts of the description that were unclear in the original version. We have therefore added a paragraph in the section "A metric to study the memorization of fuzzy duplicates" explaining the connection between MIA AUC, memorization, and the effect of duplicates on it.

We have also clarified the definition of ν_{eq} as the value of ν for which ϕ_{ν} best approximates $\tilde{\phi}$.

It remains unclear to us, over **which models** the fuzzy duplicates with token insertions and shuffling actually are tested? This is not specified in text or captions. The model employed could also be mentioned in the captions of the figures to improve readability.

We agree that this should have been made clearer in the manuscript. The experiments related to token insertions and shuffling have been conducted with GPT-Neo 1.3B as target model. We have updated captions for all relevant figures in the main text as well as in *Methods* to clarify which model has been used.

Please clarify how the **baselines of the fuzzy duplicates with token insertions are defined**. On page 4, what does it mean that "n-grams are randomly scattered" and how is the insertion of infinite elements defined? The comparison to the baseline in the shuffling experiment is also not clear.

We thank the reviewer for pointing this out. The baseline for both the experiments (Figures 2a, 2b) corresponds to the case where n -grams are learned in isolation, without any associations between them. For this we select the injection position for each n -gram independently, making it very unlikely for any two n -grams from the same canary to appear within one context window during training.

If ρ exceeds this baseline, it indicates that the target model has learned some connections between the n -grams, piecing together information across insertions and reorderings, contributing to the memorization of the reference canaries.

We have now clarified the meaning of this baseline for both the token insertion and shuffling experiments in the manuscript.

In the comparison to paraphrasing, the ρ in Table 1 is compared to that of Figure 3. However, as far as we understand, the figure and table are conducted under different models. Does this ensure a fair comparison?

We thank the reviewer for catching this and apologize for not making it clear in the original manuscript. Both reported results (the values of the exact duplicate equivalent ρ) correspond to using the same target model, namely GPT-Neo 1.3B. To avoid confusion, we now clarify that the different models mentioned in Table 1 refer to the instruction-tuned models used to paraphrase the reference canaries, rather than to the target model itself. We have updated the caption of Table 1 accordingly.

Some minor suggestions and questions:

- While the experiment studying syntactic versus semantic memorization is interesting, is the conclusion that it is predominantly syntactic actually "surprising"? Isn't it well understood that machine learning actually learns and produces preexistent patterns, but does not understand/learn the semantic meaning of these patterns?

Thank you for this comment. We believe this result may be surprising to some in the community, but agree that it also may be expected by others.

On the one hand, recent advances in LLMs suggest they go far beyond simple pattern matching, demonstrating increasingly abstract understanding and strong reasoning capabilities. Prior work has reported '*emergent capabilities*' [A1], with models showing near human-level performance across domains [A2] and advanced reasoning abilities [A3, A4, A5], and even achieving a gold medal in the International Mathematical Olympiad [A6]. From this perspective, it may seem surprising that LLMs do not memorize more substantially across semantically coherent fuzzy duplicates.

On the other hand, Bender et al. [A7] described language models as stochastic parrots, i.e. "*a system for haphazardly stitching together sequences of linguistic forms it has observed in its vast training data, according to probabilistic information about how they combine, but without any reference to meaning.*" Notable recent work echoes similar skepticism on recently reported capabilities, suggesting that emergent abilities may be artifacts of model evaluation [A8] and that current models' reasoning breaks down as problem complexity increases [A9].

- In general, the section on "Quantifying a mosaic memory" is a bit hard to follow and could benefit from revision

As also mentioned above, we have added further clarifications to this section in the main text, and now more explicitly refer to the relevant details in the Methods.

- Please be sure to indicate in the introduction that you are working with 100 tokens. Otherwise, sentences such as "Levenshtein distance 10, for instance, represents just 10% of tokens" may confuse.

Addressed.

- Note that "baseline" is repeated in "(against the baseline baseline of $\rho=0.321$)".

Addressed.

- The labelling of Figure 5 would benefit from improvement. Could the actual n-gram sizes be included just next to the lines (for example, left to where they finish around $x=50$)? The color grading makes them very hard to distinguish. In addition, the in-text references to specific curves may not be immediately clear, especially since the first line from the bottom is almost invisible except for the slight increase near the end. It is easy then to miss that curve when finding the "fifth line from the bottom", which the text highlights. In addition, wouldn't the curve for $n=25$ mentioned on page 12 be actually the one for $n=20$? It seems that the curves are in 10-step intervals, but it is hard to confirm due to the similar color tones.

We thank the reviewers for the suggestion. We have now labeled individual lines with the corresponding values of n . We have also corrected the reference to the $n=20$ curve on page 12 – the curves are indeed in 10-step intervals.

- Please note that there are minor misprints in some references, such as 49 and 61.

Addressed.

Additional references used to respond to reviewer #1

[A1] Wei, J., Tay, Y., Bommasani, R., Raffel, C., Zoph, B., Borgeaud, S., ... & Fedus, W. Emergent Abilities of Large Language Models. Transactions on Machine Learning Research.

[A2] Bubeck, S., Chandrasekaran, V., Eldan, R., Gehrke, J., Horvitz, E., Kamar, E., ... & Zhang, Y. (2023). Sparks of artificial general intelligence: Early experiments with gpt-4. arXiv preprint arXiv:2303.12712.

[A3] Kojima, T., Gu, S. S., Reid, M., Matsuo, Y., & Iwasawa, Y. (2022). Large language models are zero-shot reasoners. Advances in neural information processing systems, 35, 22199-22213.

[A4] Wei, J., Wang, X., Schuurmans, D., Bosma, M., Xia, F., Chi, E., ... & Zhou, D. (2022). Chain-of-thought prompting elicits reasoning in large language models. Advances in neural information processing systems, 35, 24824-24837.

[A5] Guo, D., Yang, D., Zhang, H., Song, J., Wang, P., Zhu, Q., ... & Tan, Y. (2025). DeepSeek-R1 incentivizes reasoning in LLMs through reinforcement learning. Nature, 645(8081), 633-638.

[A6] Luong, T., & Lockhart, E. (2025, July 21). *Advanced version of Gemini with Deep Think officially achieves gold-medal standard at the International Mathematical Olympiad*. Google DeepMind.

[A7] Bender, E. M., Gebru, T., McMillan-Major, A., & Shmitchell, S. (2021, March). On the dangers of stochastic parrots: Can language models be too big? 🦜. In Proceedings of the 2021 ACM conference on fairness, accountability, and transparency (pp. 610-623).

[A8] Schaeffer, R., Miranda, B., & Koyejo, S. (2023). Are emergent abilities of large language models a mirage?. Advances in neural information processing systems, 36, 55565-55581.

[A9] Shojaee, P., Mirzadeh, I., Alizadeh, K., Horton, M., Bengio, S., & Farajtabar, M. (2025). The illusion of thinking: Understanding the strengths and limitations of reasoning models via the lens of problem complexity. arXiv preprint arXiv:2506.06941.

[A10] Grosse, R., Bae, J., Anil, C., Elhage, N., Tamkin, A., Tajdini, A., ... & Bowman, S. R. (2023). Studying large language model generalization with influence functions. *arXiv preprint arXiv:2308.03296*.

Reviewer #4 (Remarks to the Author and Authors' Response)

We thank the reviewer for their time spent on the manuscript and for their constructive and thoughtful feedback, which helped us improve the manuscript.

Key results

The motivation of the study is to **reduce sensitive information leakage of LLMs due to fuzzy duplications** in the training dataset. The “sensitive information” is simulated by synthetic sequence generated using an LLM, which the authors refer as “canaries”.

We would like to clarify that the notion of canaries, i.e. unique and specifically crafted sequences injected into language model training data to study memorization/susceptibility to MIAs, has been used extensively in the privacy literature [14, 22, 46, 47, 48, 49, 50]. In this work, we adopt the technique to synthetically generate canaries as used by Meeus et al. [22].

“Memorization” is measured using Membership Inference Attacks (MIAs). Specifically, canaries along with their ‘fuzzy’ duplicates (constructed by replacement, insertion, paraphrases, etc.) are used for additional training a pre-trained LLM. Their (log) probabilities after training are compared to canaries that are not used in training.

The main result of the paper is measuring the contribution of the presence of fuzzy duplicates to the memorization of the canaries. For this they introduce a relative measure of the memorization, exact duplicate equivalent (ρ), which evaluate how many fuzzy duplications are equivalent to one exact duplication. The authors found that, depending on the degree of fuzziness (e.g. the number of replacements, Levenshtein distance), a small number of fuzzy repetitions can have a similar effect (e.g. sequence probability) as having one exact repetition. In the later parts of the study, the authors examine a popular deduplicated dataset and show that it still contains many fuzzy duplicated sequences. Using a more aggressive deduplication procedure, they can further remove fuzzy duplications in that dataset.

Validity

The experiments are mostly based on the existing methods, such as Membership Inference Attacks, Meeus, M (2024). The methods and results are valid. **However, the authors may overinterpret the results out of the scope of the experiments. The actual implication of the results is vague and complicated. There is almost no statistical test on any results.**

We agree on the importance of demonstrating statistical robustness and reproducibility, as highlighted by the reviewer.

We now report, in all of our main results (Figures 1–3, Table 1), the mean and standard deviation across five independent runs, each with randomness in both canary generation and model initialization. These show the results and findings to be consistent across runs. For experiments using real data (Figure 4b), when estimating the average number of fuzzy duplicates in SlimPajama, we now report the standard error of the mean.

The main measure, ρ , is used to make most of the claims in the manuscript. However, ρ measures the effect of fuzzy duplicates relative to exact duplicates. ***Though not claimed directly, the author imply that higher ρ is undesirable, because higher ρ means the fuzzy duplicates are as “detrimental” as exact duplicate.***

We would like to clarify that our goal in this work is not to make a normative judgment about memorization in language models. As discussed in the Introduction, memorization can be beneficial, for instance by improving generalization [8–10] and supporting the encoding of factual knowledge [11, 12]. However, it can also raise concerns when it involves the retention of personal or confidential information [14–19], copyright-protected material [20–22], or benchmark data [23–29]. In this study, we aim to understand the mechanisms through which memorization occurs and to place them in the context of widely used deduplication practices employed by model developers to mitigate such effects.

However, it is not clear how bad the exact duplicates are. Thus, it is ***difficult to relate ρ to impact on realistic use cases.*** For example, ρ does not tell how many (fuzzy) duplicates are sufficient to induce unsafe information leakage and with what probability.

We thank the reviewer for raising this point. We agree that the exact effect of duplicates on memorization depends on the specific training setup (e.g., model size, dataset size, hyperparameters).

We motivate our work based on two key findings from prior research. First, training data can, under certain conditions, be extracted from LLMs. Carlini et al. (2021) [15] demonstrated this for GPT-2, and subsequent work by Nasr et al. (2023) [57] showed that ChatGPT (gpt-3.5-turbo) is similarly vulnerable. Other studies focusing on private [18, 19] and copyright-protected [20, 21] content further confirm that the threat of training data extraction is realistic and observable in real-world models. Second, prior research has shown that exact duplication of training samples increases susceptibility to extraction [30, 32] and membership inference [22] attacks. Taken together, these findings motivate our choice to measure the impact of fuzzy duplicates relative to exact duplicates, as the latter have been empirically linked to higher privacy risk in realistic settings.

Moreover, *exact deduplication practices* are widely applied, not only to mitigate these privacy concerns [30], but also to enhance training efficiency [31, 35–39] and avoid benchmark contamination [40–44]. We thus believe it valuable to contextualize the effect of fuzzy duplicates in relation to these established practices.

It is also unclear whether removing duplication can eliminates information leakage, given that “personal” information which is a main topic of interest in this context, is typically not widely known (repeated in the training dataset).

We thank the reviewer for raising this point. We agree that Personally Identifiable Information (PII), while a major concern in the context of memorization, might indeed be less likely to be highly repeated due to its nature. However, this does not eliminate the risk entirely – some degree of repetition can still occur. For instance, medical records may appear across multiple databases, or emails and physical addresses may be stored on different websites. Given the sensitive nature of such data and the sheer scale of modern training corpora, even rare instances of memorization are cause for concern.

Beyond LLM pretraining, models are also increasingly fine-tuned on smaller, domain-specific datasets e.g. from the medical, legal or financial domain or human-chatbot interactions. Fuzzy duplicates are also likely to occur in this setting (e.g. the same sensitive information appearing in different files), and the risk of not addressing the effect of fuzzy duplicates/memorization in such cases might be larger than in pretraining.

Moreover, personal information is not the only domain where memorization poses a problem. The memorization of copyright-protected content is increasingly recognized as a significant issue, with recent work showing that texts such as *Harry Potter* and *1984* are almost fully memorized by LLaMA-3.1 [A10]. In this context, both exact and fuzzy duplication are likely to occur, further amplifying the risk.

Based on the results in Fig. 1, the authors claim that “fuzzy duplicates contribute significantly to memorization.” This is a questionable claim. ***The claim comes from the observation that on all “number of tokens R replaced” except 100 the rho is larger than 0. This observation could be trivial.*** Two points on the plot ((0, 1.0) and (100, 0)) are there by construction. Simply connecting the two points result in all middle values larger than 0.

We would like to clarify that our key finding is not merely that all values of ρ remain above 0, but the quantitative characterization of the impact of a wide range of fuzzy duplicates on memorization. The key contribution lies in the *shape* of the curve and what it reveals about how memorization behaves as perturbations increase. If memorization were predominantly verbatim, as commonly assumed in prior work, the curve would drop sharply once tokens are replaced, reflecting a rapid loss of memorization. Instead, we observe a gradual decay, showing that even substantially altered sequences continue to reinforce memorization.

We further quantify this effect across different types of fuzzy duplicates (replacement, insertion, and shuffling) and across a wide spectrum of perturbation levels (e.g., R ranging from 1 to 75). For instance, as shown in Figure 1, ten fuzzy duplicates with 20% of their tokens replaced yield a memorization level equivalent to repeating the canary four times exactly. We believe this systematic quantification along this entire spectrum of perturbations provides non-trivial and informative insights about how partial overlaps reinforce memorization.

More quantitative claim “In practice this means that having two fuzzy duplicates in the training data with 10% of the original tokens replaced yields higher memorization in the than one exact repetition” is technically true, but ***it is not clear what the memorization effect of one exact repetition is.***

We thank the reviewer for raising this point. We agree that it is helpful to provide clearer context for the effect of exact repetitions. We have therefore added an explicit reference to

Section M5, which reports the relationship between exact repetitions and vulnerability to MIAs. As shown in Figure M1, the MIA AUC increases from 0.58 for one exact repetition to 0.85 for five repetitions and 0.95 for ten repetitions, illustrating how repeated exposure progressively amplifies memorization.

Another issue is that the main experiments are about finetuning pretrained LLMs (A minor terminology suggest: the authors describe their training procedure as “further pretraining” of a pretrained model. However, “Continued pretraining” is a more prevalent name than “further pretraining” by convention.).

We agree that continued pretraining is a more widely used term, and have now adopted this naming convention throughout the paper.

However, the role of different finetuning protocols is not discussed. For example, the learning rate could affect ρ .

We agree with the reviewer that different fine-tuning protocols can influence memorization. We therefore examine in Section M6 how varying the learning rate affects ρ (Figure M3), along with additional analyses across different model families, model sizes, and canary properties. These results show that our findings are consistent across a range of training configurations.

More importantly, it is unclear whether memorization of sequences introduced in this fine-tuning is similar to memorization of sequences introduced in the complex and long pre-training process.

We thank the reviewer for raising this important point. We agree that this is indeed a limitation of our study. Our experiments focus on memorization during continued pretraining, and it remains an open question how these dynamics might differ during full-scale pretraining.

However, we chose this setup as continued pretraining closely approximates later stages of pretraining while remaining practical and methodologically tractable. In particular:

1. Conceptual proximity to pretraining. We use the same training objective and loss function as in pretraining (i.e. next-token prediction), rather than post-training methods such as reinforcement learning, making our setup closely resemble the late stages of pretraining.

2. Computational feasibility. Our experiments involve over 100 model training runs across multiple setups and types of fuzzy duplicates. Performing such an analysis at full pretraining scale would be prohibitively expensive, especially within an academic budget.

3. Prior work. We note that many prior works have studied MIAs in a continued pretraining setting for these same reasons [17, 18, A5–A7].

4. Control over data distributions. Evaluating MIAs requires precise control over which samples are included (members) and excluded (non-members), ensuring both are drawn from the same distribution. Prior work has evaluated MIAs at the pretraining level using data

collected *post-hoc*, i.e. before and after the training data cutoff date [52, 72]. Yet, this approach introduces a severe distribution shift, which was recently shown to yield misleading conclusions [A1-A4]. Especially the fine-grained control required to study the wide range of fuzzy duplicates in this work, is for us only achievable in a continued pretraining setup.

5. Focus on relative effects. Our analysis quantifies the *relative* impact of fuzzy versus exact duplicates, abstracting away from the absolute level of memorization – which indeed may differ between early and late-stage pretraining. We hypothesize that the relative effects are less sensitive to the training stage, though we leave a detailed investigation of this question to future work.

Significance

Understanding the memory of LLMs is a fundamentally important topic. A main claim of the manuscript is that the fuzzy duplicates in the training data can lead to undesired memorization. Additional interesting result is the observation that syntactic more than semantic, similarity determines the efficacy of fuzzy duplicates.

However,

(1) The fact that memorization can be enhanced by inexact repetition is not surprising, and is **clearly demonstrated by recent work on paraphrases**. The general significance of the quantitative measure adopted here is questionable, as explained above.

We thank the reviewer for highlighting this connection to Allen-Zhu et al. [A8].

We view our work as complementary to this work [A8], as it focuses on a different aspect of inexact duplication. Their study focuses on *knowledge augmentation*, i.e. adding paraphrases, sentence shuffles, or translations during pretraining, to improve *generalization* and *factual recall* after instruction tuning. In contrast, our goal is to understand how fuzzy duplicates affect *verbatim sequence memorization*. We (i) quantify this effect relative to the effect of exact duplicates across many modification types, (ii) find that primarily syntactic (rather than semantic) modifications drive this effect and (iii) connect this to commonly deployed deduplication mechanisms.

To highlight the contrast between the two works, Allen-Zhu et al. show that including inexact duplicates can reduce reliance on verbatim recall, helping the model transition from merely reproducing input sequences to reasoning about underlying facts. In contrast, our results show that from a membership-inference perspective, the inclusion of inexact duplicates actually strengthens memorization.

While, following results on knowledge extraction from [A8], one might find it intuitive that inexact repetition contributes to memorization, our work provides a systematic quantitative characterization of this effect, revealing clear differences across duplicate types and their implications for data deduplication.

(2) In the context of data privacy and copyright protection, memorization is undesired. However, in a broader sense, memorization is often thought to be a desired property of an LLM. Remembering more factual details can reduce model hallucination. **The manuscript does not explain the tradeoff of the two.** The “canaries” which are used as a probe are

neutral sequences (i.e. general natural language content generated by an LLM with T=1). It is unclear whether they are a good model of desired or undesired memorized items.

We agree and intentionally did not assign normative judgments to memorization. Prior work (e.g., [9]) shows memorization can be beneficial (e.g. supporting generalization, recalling facts) or undesirable (privacy/copyright leakage). As also noted in [A8], an LLM remembering the birthday of Abraham Lincoln is desirable, while recalling that of a non-public individual is not. Our aim is to study and quantify the mechanism: how fuzzy duplicates affect memorization, irrespective of whether the memorized content is ultimately helpful or harmful. To that end, we primarily use neutral canaries, not because they model all desirable/undesirable items, but because they approximate representative training data. We vary the temperature with which the canaries have been generated in Section M6 and Figure M6 – and conclude that our findings hold across a range of different canary generation scenarios.

Without making any normative judgements, we observe that many works deduplicate data in practice for different reasons: to reduce privacy risks, improve efficiency, and ensure fair model evaluation. Our results show that exact deduplication often fails to remove impactful fuzzy duplicates, which continue to influence model memorization. Hence, our findings are relevant to deduplication practices regardless of their specific motivation.

There are well-recognized studies showing paraphrases are necessary for generalization (for example Zeyuan Allen-Zhu 2024 part 3.1). The authors do not discuss how they fuzzy duplicates and paraphrases help generalization but only discussed their role on “memorization”.

We indeed focus on how fuzzy duplicates contribute to memorization rather than to generalization. This perspective led to findings such as the quantitative comparison of fuzzy and exact duplicates, and the observation that syntactic, rather than semantic, overlap most strongly drives memorization. Building up on the results from Allen-Zhu et al. and studying how fuzzy duplicates affect *generalization* (e.g. in terms of knowledge extraction, performance on unseen benchmarks) would be an interesting direction for future research beyond the scope of this work.

Data and methodology

There is no issue on the data. The methods are clearly explained in the supplementary. The LLM models used in the experiments are two generations behind, but still acceptable.

Analytical approach

There is no statistical analysis except standard deviation in the tables. Adding statistical analysis to most of the plots will be helpful.

Suggested improvements

The authors state that the deduplication process of *SlimPajama fails to even remove exact duplicates. However, the authors did not discuss potential reasons of the failure.* It is surprising that a widely-used dataset fails so evidently. *Some examples would be helpful to understand the failure.* One potential reason I can think of is that sequence

length used in the SlimPajama deduplication and in this manuscript are different, so that there are shorter sequences that are exactly the same.

We thank the reviewer for this comment – this is indeed one of the more interesting aspects of our findings, and we are happy to clarify it further. The key distinction, that we highlight in the paper, is that SlimPajama was subject to **document-level** deduplication, removing documents with a 13-gram Jaccard similarity above 0.8, while we look for **sequence-level duplicates** that persist within or across documents. To address the reviewer’s feedback we expand and clarify this distinction in the paper. We also elaborate on this in the *Discussion* section.

We additionally include example sequences with their fuzzy duplicates in Table M8, illustrating potential mechanisms through which such duplicates can emerge in the dataset.

As the authors state in the section of “deduplication”, there is a trade-off in the choice of deduplication approaches. However, the authors only show the effectiveness of sample-level deduplication (Fig. 5). The result in Fig. 5 is as expected but does not answer important questions like what the optimal deduplication procedure is.

Our goal with Figure 5 is to quantify how much current exact deduplication practices miss in terms of fuzzy duplicates, rather than to propose an optimal deduplication strategy. As discussed in the *Discussion* section, the optimal level of deduplication inherently involves application- and scale-specific trade-offs. For example, when deduplicating fine-tuning data in sensitive domains (e.g., medical records) or when a large AI lab seeks to rigorously decontaminate evaluation benchmarks, one may prioritize stricter exact deduplication thresholds (i.e., smaller n). In such settings, the datasets are often relatively small, making it even feasible to use more computationally intensive fuzzy deduplication methods such as those based on Levenshtein distance.

In contrast, for large-scale public pretraining, where deduplication primarily targets training efficiency and model quality, both the computational cost of sequence-level matching (e.g., Levenshtein across trillions of tokens) and the data loss incurred by using small n -values can outweigh the benefits. In such cases, allowing some fuzzy duplicates to remain may be a reasonable compromise.

We therefore refrain from prescribing a universal “best” deduplication strategy, as the appropriate balance depends strongly on the scale and purpose of the application. A comprehensive exploration of these trade-offs would require substantial computational resources, and any resulting conclusions would likely depend on the specific use case; we therefore leave such an analysis to future work. We outline key considerations affecting this trade-off in the *Discussion* section while deliberately avoiding prescriptive recommendations of a universally optimal strategy.

Additional references used to respond to reviewer #4

[A1] Duan, M., Suri, A., Mireshghallah, N., Min, S., Shi, W., Zettlemoyer, L., ... & Hajishirzi, H. Do Membership Inference Attacks Work on Large Language Models?. In *First Conference on Language Modeling*.

[A2] Meeus, M., Shilov, I., Jain, S., Faysse, M., Rei, M., & de Montjoye, Y. A. (2025, April). Sok: Membership inference attacks on llms are rushing nowhere (and how to fix it). In *2025 IEEE Conference on Secure and Trustworthy Machine Learning (SaTML)* (pp. 385-401). IEEE.

[A3] Maini, P., Jia, H., Papernot, N., & Dziedzic, A. (2024). LLM Dataset Inference: Did you train on my dataset?. *Advances in Neural Information Processing Systems*, 37, 124069-124092.

[A4] Das, D., Zhang, J., & Trantèr, F. (2025, May). Blind baselines beat membership inference attacks for foundation models. In *2025 IEEE Security and Privacy Workshops (SPW)* (pp. 118-125). IEEE.

[A5] Mattern, J., Mireshghallah, F., Jin, Z., Schoelkopf, B., Sachan, M., & Berg-Kirkpatrick, T. (2023, July). Membership Inference Attacks against Language Models via Neighbourhood Comparison. In *Findings of the Association for Computational Linguistics: ACL 2023* (pp. 11330-11343).

[A6] Zeng, S., Li, Y., Ren, J., Liu, Y., Xu, H., He, P., ... & Yin, D. (2024, August). Exploring Memorization in Fine-tuned Language Models. In *Proceedings of the 62nd Annual Meeting of the Association for Computational Linguistics (Volume 1: Long Papers)* (pp. 3917-3948).

[A7] Dentan, J., Buscaldi, D., Shabou, A., & Vanier, S. (2024). Predicting and analyzing memorization within fine-tuned Large Language Models. *arXiv preprint arXiv:2409.18858*.

[A8] Allen-Zhu, Z., & Li, Y. (2024, July). Physics of language models: part 3.1, knowledge storage and extraction. In *Proceedings of the 41st International Conference on Machine Learning* (pp. 1067-1077).

[A9] Cooper, A. F., Gokaslan, A., Ahmed, A., Cyphert, A. B., De Sa, C., Lemley, M. A., ... & Liang, P. (2025). Extracting memorized pieces of (copyrighted) books from open-weight language models. *arXiv preprint arXiv:2505.12546*.

Reviewer #6 (Remarks to the Author and Authors' Response)

We thank the reviewer for their time spent on the manuscript and for their constructive and thoughtful feedback.

There has been a long line of work analyzing the memorization of training examples in the parameters of neural networks, which in recent years has been extended to large language models (LLMs). In LLMs, such memorization typically occurs due to duplicates in the training data, i.e., the same document or string being present multiple times. While previous work has predominantly looked at memorization resulting from exact duplicates, the present paper shows that memorization happens even if the duplicates are fuzzy, for example if some of the tokens in them are swapped or the strings are rephrased versions of the same string. By carefully choosing the fuzzy duplicates to be at varying semantic distances from the original string, the authors show that memorization is rather syntactic than semantic: operations that increase semantic distance have a smaller effect on memorization than those that increase syntactic distance. For measuring memorization, the authors use the ROC AUC of different membership inference attacks (MIAs) from the literature and introduce the metric ρ to relate the memorization of fuzzy duplicates to the memorization of exact duplicates.

Given the potential impacts of the memorization of training data (leakage of confidential information, copyright implications, etc.), mitigation of memorization and of membership inference and extraction attacks is an important problem. The authors show that mitigation techniques relying on exact deduplication do not suffice. ***This is an important contribution with potential real-world impacts, and is worth being published. On the more foundational side, the findings that point towards models remembering syntactically rather than semantically are an interesting contribution to ongoing discussions about the internal workings of LLMs and their abstraction abilities.***

On the methodological side, the paper is sound. The authors perform many ablation studies, which are particularly crucial for such synthetic experiments. These ablations largely show the robustness of the findings. The main methodological contribution of the paper is the duplicate equivalent metric ρ , which measures how many exact duplicates a fuzzy duplicate corresponds to. ***This is an intuitive metric because it is mostly independent of the absolute level of memorization, and seems useful for future work.*** The other methodological contributions are all quite natural, but not particularly innovative.

We would like to thank the reviewer for their positive assessment and for recognizing the methodological soundness of our work.

One methodological criticism that I have: generating reference canaries via sampling from an LLM could lead to generating sequences that are memorized by that LLM, which are in turn likely to occur multiple times in the training data because they are memorized. The authors use Llama 2 as the reference model, which they do not use for evaluating memorization, but it is still likely that at least some of its training data overlaps with that of the models that they evaluate. Following this logic, it might be that the canaries are sequences that are already duplicated in the models' training data, thus distorting the ρ values. This criticism is partly addressed through ablation studies with higher sampling temperatures for the reference canaries.

We thank the reviewer for raising this subtle but important point. While it is unlikely that any of our synthetically generated 100-token canaries (sampled with temperature 1) appear verbatim in the pretraining data, we agree that, given the immense scale of modern datasets, it is difficult to entirely rule out the presence of some exact or fuzzy duplicates.

We would, however, like to clarify that such potential overlap does not compromise the validity of our results. The experimental design was intentionally constructed to remain robust even under this possibility.

First, the MIAs are conducted specifically against the *continued pretraining stage*, where we explicitly control which canaries are injected. Because both member and non-member canaries are generated using the same process, they are equally likely to have overlaps with the pretraining corpus. Any such, overlap would therefore affect both groups symmetrically and not bias the evaluation.

Second, although such overlap might influence the *absolute* degree of memorization, our main analysis concerns *relative* memorization – captured by ρ – which quantifies how fuzzy duplicates compare to exact duplicates within the same setup. As shown in Section M6, our

results remain consistent across a wide range of hyperparameters that modulate overall memorization strength, demonstrating that ρ is stable under such conditions. This includes the case in which canaries are generated with a temperature up to 5, where overlap with pretraining data becomes exceedingly unlikely.

There are two additional types of experiments that would make the paper stronger.

The first is an experiment that brings it all together. The authors show that memorization occurs with synthetic canaries and fuzzy duplicates, and show that such fuzzy duplicates occur in the LLM training dataset SlimPajama. A very natural next step, and the final proof that memorization due to fuzzy duplicates is a real problem, would be training an LLM on (a subset of) a dataset such as SlimPajama and measuring the memorization of strings with naturally occurring fuzzy duplicates. Even though the algorithms used for generating the canaries and for producing fuzzy duplicates used by the authors are reasonable, the resulting sequences still remain synthetic, and real-world text and fuzzy duplicates will look different. The authors already detect fuzzy duplicates and estimate their frequency in SlimPajama, so running their training and MIA attacks with this data doesn't seem like a large amount of effort. This is just my outside perspective, though, and I might be overlooking challenges with such an experiment.

Thank you for this great suggestion. We agree that it is valuable to confirm that our findings for synthetic reference canaries, and their constructed fuzzy duplicates, also align with real-world fuzzy duplicates.

Following the reviewer's suggestion, we select reference canaries and their fuzzy duplicates from SlimPajama within a certain Levenshtein distance range and repeat the computation of the exact duplicate equivalent ρ for these real-world fuzzy duplicates. We now visualize the results together with our other results in Figure 4(a). We find that ρ decreases with Levenshtein distance similarly to the trend observed for our other experiments, confirming that our findings generalize to real-world fuzzy duplicates. Further details are provided in Section M12.

Many of the risks of memorization only materialize if it is not only possible to determine that a string was used during training, but if (parts of) it can be **extracted** from the model. Carlini et al. define the concept of extractable memorization [1,2]: A string s is extractable from a model if one can construct a prompt so that the model generates s . See also Lukas et al. for the case of PII [3], and the work connecting membership and attribute inference [4,5]. In addition to MIA attacks, ***the authors could have also tested extraction attacks to see whether those also still work in the presence of only fuzzy duplicates.*** In the absence of that, it would still be good if they could (briefly) comment on the difference between membership inference and extraction.

Following the reviewer's suggestion, we additionally study how fuzzy duplicates contribute to the extraction risk of the reference canaries. In Section M8, we measure the extractability of reference canaries from GPT-Neo-1.3B and GPT-Neo-2.7B trained on datasets containing exact and fuzzy duplicates. We prompt the trained target model with a prefix p from the reference canary and, in line with Ippolito et al. [33], measure approximate extraction risk using BLEU and Levenshtein similarity between the ground truth and the generation (greedy

decoding). We consider prefix lengths of 50 and 75 tokens, and fuzzy duplicates obtained by making R replacements as used throughout the paper.

First, we find that approximate extraction metrics increase with more *exact* repetitions, e.g. for a prefix of 75 tokens, Levenshtein distance between the ground truth and model's output is 0.15 for $n_{dup}=1$ and 0.32 for $n_{dup}=10$.

Next, consistent with our findings on MIA vulnerability, we find that *fuzzy duplicates* contribute substantially to the approximate extraction risk of the reference canaries. Across all setups, both BLEU and Levenshtein similarity remain substantially above the baseline for $n_{dup}=1$ repetition, and approach the level observed for $n_{dup}=10$ for lower values of R (Table M5). Additional results and discussion are provided in Section M8 and refer to these results in the Results section, under "*LLMs have a mosaic memory*".

The first of those two proposed (series of) experiments is more important to me than the second, which can rather be seen as a suggestion for future work. In case the first experiment would be significantly more challenging and a larger amount of work than I am thinking right now, I would suggest accepting the paper even without it.

Lastly, I have a few questions for the authors:

1. For the token replacement via RoBERTa, how do you deal with the fact that this model has a different tokenizer than some of the models that you evaluate memorization on? E.g., if you want to replace one token, but the token returned by RoBERTa is actually tokenized into two tokens by the tokenizer of the model that you evaluate? This would be good to clarify in the paper.

Many thanks for raising this question; it is indeed a subtle technical detail we had to address. In our implementation, we always sample exactly one token from the MLM's vocabulary, which may correspond to multiple tokens under the target tokenizer – which means that the resulting canary can sometimes have slightly more than 100 tokens. We have clarified this in section M2 in *Methods*.

2. "the respective p also rise from 1.99 to 2.51 and 3.72, respectively" (p.8, top). Those seem to be the wrong numbers, as p should be between 0 and 1. Do you mean the numbers from Table 1, i.e., 0.11, 0.17, 0.30?

We thank the reviewer for pointing that out correctly and we have now addressed this.

3. In M8 you write that you did not control the sampling parameters for GPT-4o. Was it not possible to set the temperature back when you used the API for these experiments, or does the statement only refer to the probability sum?

Thanks for this question. While we were indeed able to specify the sampling parameters when querying GPT-4o, we chose to use the default ones for OpenAI's API. We have clarified this in section M8 in *Methods*.

[1] Carlini et al. Extracting Training Data from Large Language Models. 2023. USENIX

[2] Nasr et al. Scalable Extraction of Training Data from (Production) Language Models. 2023. arXiv

[3] Lukas et al. Analyzing Leakage of Personally Identifiable Information in Language Models. 2023. SP

[4] Yeom et al. Privacy Risk in Machine Learning: Analyzing the Connection to Overfitting. 2018. CSF

[5] Salem et al. SoK: Let the Privacy Games Begin! A Unified Treatment of Data Inference Privacy in Machine Learning. 2023. SP(Remarks on code availability)

The Mosaic Memory of Large Language Models (NCOMMS-25-29287):

Response to reviewers (Second Revision, December 2025)

We thank the reviewers for their valuable feedback and the time they dedicated to our manuscript. In this document, we respond to each reviewer's comments, reproducing their comments verbatim and providing our responses in blue.

Reviewer #1 (Remarks to the Author):

We would like to thank the authors for their hard work in improving the manuscript. We believe that all of the comments and questions raised in our review have been addressed in this new version or in the author's response.

Overall, we are grateful for the effort made to make the paper more accessible and for all the additional descriptions and specifications provided. We believe these improvements have substantially enhanced the paper. Nevertheless, we have a few final minor comments about this new version.

We thank the reviewer for their positive assessment and are pleased that our revisions have addressed their concerns. We appreciate their constructive engagement throughout the review process.

Firstly, we appreciate the detailed answers given in the Author's Response, but we believe that some have not been fully addressed in the final version of the manuscript itself. For example, the fact that " p captures how fuzzy duplicates influence memorization regardless of how inherently "memorable" a sequence is", or that " ν_{eq} " is "the value of ν for which ϕ_{ν} best approximates $\tilde{\phi}$ " are very clear in the Author's response, but still not as clear in the newer draft (in particular, note that the latter is much clearer in the author's response than in the new version "the value of ν for which $\tilde{\phi} \approx \phi_{\nu_{eq}}$ ", which is somewhat ambiguous). Ideally, we would like the authors to incorporate the specifications from the Author's Response into the text.

We thank the reviewer for this helpful suggestion. We agree that the clarifications provided in our previous Author's Response should be incorporated directly into the manuscript to ensure the paper is self-contained and clear.

We have now revised the relevant passages in the main text. Specifically:

1. We explain the intuition behind the design of our metric (ρ) in the beginning of the relevant section (page 3, "A metric to study the memorization of fuzzy duplicates").
2. We expand paragraphs defining ν_{eq} , providing additional context behind our design (page 4, "A metric to study the memorization of fuzzy duplicates").

Similarly, regarding our comment on the "surprisingness" of the results, we appreciate the authors' explanation and acknowledge that the issue is more contentious than we previously thought. We believe that the answer provided in the authors' response is important, and we would like it to be mentioned or discussed in the paper as well. This would slightly increase transparency with regards to the current knowledge of the community.

We thank the reviewer for raising this point. We agree that clarifying why the syntactic over semantic memorization result is surprising improves the transparency of the manuscript. Following this suggestion, we have expanded the discussion in the main text (both on page 2, "Introduction" and page 6, "The mosaic memory is syntactic rather than semantic") to explicitly place this result in the context of the increasingly strong semantic capabilities reported for modern language models (e.g., reasoning, problem solving, code generation). By contrasting these widely observed abilities, often interpreted as requiring semantic abstraction, with our finding that memorization across fuzzy duplicates is nevertheless driven primarily by token-level overlap, we hope to better motivate why this result is non-trivial and informative for the community.

We also think that the additional references to the appendices will help readers, as we can now see that they address some of our initial questions.

We thank the reviewer for this suggestion. We have now added more explicit references to the appendix, specifically calling out our experiments with varying canary sampling temperature.

However, we noticed that the rewriting has introduced a few terms into the main body that might require explaining. For instance, the term "temperature" in the added text "setting the temperature $T=1$ to mimic realistic text" is not explained in the main body.

We thank the reviewer for pointing this out. We have now added a brief explanation of temperature in the main text (page 3, "A metric to study the memorization of fuzzy duplicates"), clarifying that it controls the randomness of token sampling and that $T=1$ uses the model's natural probability distribution to produce coherent, yet varied text.

We also realise that our comment on the canaries chosen by the same distribution was incorrect. We appreciate the authors helping us to better understand, and we agree with their response to our comment.

Finally, we noticed some typos:

- * The T in temperature (pg 3) should be in \mathcal{T} in concordance with the appendices and to avoid confusion with the tokenizer T .
- * Missing spacing after "indicating stronger memorization" (pg 3).
- * Missing words in "yields higher memorization in the than one exact repetition" (pg 4).

We thank the reviewer for listing these typos and have now addressed them.

We would like to thank the authors once again for their responses and corrections.

We would like to thank the reviewers for their thoughtful and constructive feedback throughout the review process. Their insightful comments and suggestions have helped us substantially improve the clarity and rigor of the manuscript. We are pleased that our revisions have addressed the concerns raised, and we are grateful for the reviewers' engagement with our work.

Reviewer #2 (Remarks to the Author):

Reviewer #4 (Remarks to the Author):

I would like to thank the authors for their detailed response to my comments. I appreciate the effort taken to address the concerns raised regarding validity, significance, and methodology.

Statistical Robustness: I appreciate the inclusion of the mean and standard deviation across independent runs for the main results.

Clarification of Scope: The authors have provided a clear distinction that their work aims to study the mechanism of memorization rather than making normative judgments of the consequences of memorization. Though I still believe the pure empirical approach could only give limited insights to the mechanism of memorization, and interesting and significant problems, such as the trade-off between data deduplication and generalization, are left to future works.

Comparison to Prior Work: I accept the distinction drawn between this study and the work of Allen-Zhu et al.

SlimPajama Deduplication: The clarification regarding the difference between document-level (Jaccard similarity) and sequence-level (Levenshtein) deduplication effectively explains why fuzzy duplicates persist in the dataset.

Overall, the revisions and additional explanations have strengthened the manuscript. I am mostly satisfied with the authors' response.

We sincerely thank the reviewer for their thoughtful and constructive engagement throughout the review process. We are pleased that our revisions have addressed their concerns, and we are grateful for the valuable suggestions that have helped us improve the manuscript considerably.

Reviewer #5 (Remarks to the Author):

Reviewer #6 (Remarks to the Author):

Creation of the canaries: There exist examples in the literature where models prompted with just the beginning-of-sequence token have produced sequences from the training data, e.g., [1, 2]. I am, however, sufficiently convinced by the experiments with higher sampling temperatures. While canaries that are already duplicates could influence the concrete value of ρ , this should not affect the trends visible in the experiments and the main conclusions. Overall, I am not too concerned about this potential issue.

Experiment with real-world fuzzy duplicates: It's great that the authors have conducted the proposed experiment. It adds a lot to the authors' argument that fuzzy duplicates should be taken seriously. The results are less clear than in the controlled experiments, but still show a trend of ρ decreasing with increasing distance between the fuzzy duplicates. In future work, it would be interesting to analyze the real-world fuzzy duplicates more closely to, e.g., identify properties of the duplicates within each bucket that might explain the unexpected jumps in ρ .

Experiment measuring extraction risk: Thank you for conducting this additional experiment. The findings are convincing and show the robustness of the impact of fuzzy duplicates on memorization and their risks beyond membership inference.

The questions that I had for the authors have been fully addressed.

I am very satisfied with the authors' response and with the changes they made to the paper and recommend acceptance.

[1] Lee et al. Deduplicating Training Data Makes Language Models Better. 2022. ACL.

[2] Carlini et al. Extracting Training Data from Large Language Models. 2021. USENIX

We are grateful to the reviewer for their careful reading of our work and their engagement with our manuscript. We are glad that we were able to address all of the reviewer's questions. In particular, we would like to express our appreciation for suggesting the experiments on extraction risk and real-world fuzzy duplicates – we believe these additions have considerably strengthened the paper. We thank the reviewer for their support and recommendation for acceptance.